# Evaluation of ECMWF IFS (version 41R1) operational model forecasts of aerosol transport by using ceilometer network measurements

Ka Lok Chan[1,2], Matthias Wiegner[1], Harald Flentje[3], Ina Mattis[3], Frank Wagner[3,5], Josef Gasteiger[4], and Alexander Geiß[1]

[1]Meteorological Institute, Ludwig-Maximilians-Universität München, München, Germany
[2]Remote Sensing Technology Institute, German Aerospace Center (DLR), Oberpfaffenhofen, Germany
[3]Department of Research and Development, Meteorological Observatory Hohenpeißenberg, German Weather Service (DWD), Hohenpeißenberg, Germany
[4]Aerosol Physics and Environmental Physics, University of Vienna, Austria
[5]now at: Institute for Meteorology and Climate Research, Karlsruhe Institute of Technology, Eggenstein-Leopoldshafen, Germany

**Correspondence:** Ka Lok Chan (ka.chan@dlr.de)

**Abstract.** In this paper, we present a comparison of model simulations of aerosol profiles with measurements of the ceilometer network operated by the German Weather Service (DWD) over 1 year from September 2015 to August 2016. The aerosol forecasts are produced by the Copernicus Atmosphere Monitoring Service (CAMS) using the aerosol module developed within the Global and regional Earth-system Monitoring using Satellite and in-situ data (GEMS) and Modelling Atmospheric Composition and Climate (MACC) projects and coupled into the European Centre for Medium-Range Weather Forecast Integrated Forecast System (ECMWF-IFS). As the model output provides mass mixing ratios of different types of aerosol whereas the ceilometers don't, it is necessary to determine a common physical quantity for the comparison. We have chosen the attenuated backscatter $\beta^*$ for this purpose. The $\beta^*$-profiles are calculated from the mass mixing ratios of the model output assuming the inherent aerosol microphysical properties. Comparison of the attenuated backscatter averaged between an altitude from 0.2 km (typical overlap range of ceilometers) and 1 km in general shows similar annual average values. However, the standard deviation of the difference between model and observation is in 8 out of 12 sites larger than the average.

To investigate possible reasons for the differences, we have examined the role of the hygroscopic growth of particles and the particle shape. Our results show that using a more recent particle growth model would result in a $\sim$22 % reduction of particle backscatter for sea salt aerosols, corresponding to a 10 %-reduction of the total backscatter signal on average. Accounting for nonspherical dust particles in the model would reduce attenuated backscatter of dust particles by $\sim$30 %. As the concentration of dust aerosol is in general very low in Germany, a significant effect on the total backscatter signal is restricted to dust episodes. In summary, consideration of both effects tends to improve the agreement between model and observations, but without leading to a perfect consistency.

In addition a strong Saharan dust event was investigated to study the agreement of the spatiotemporal distribution of particles. It was found that the arrival time of the dust layer and its vertical extent very well agree between model and ceilometer

measurements for several stations. This underlines the potential of a network of ceilometers to validate the dispersion of aerosol layers.

## 1 Introduction

Aerosols are an important constituent of the atmosphere playing a key role in the Earth's climate and weather system. They
influence the Earth's radiation budget directly by absorbing and scattering of radiation and indirectly by providing nuclei for cloud condensation. The chemical and physical properties of aerosols depend on their composition and sources. In recent decades, an increasing amount of anthropogenic aerosols is released into the atmosphere which makes it one of the largest uncertainties in assessments of climate change (IPCC, 2012). Numerous studies have been conducted in recent decades to investigate the relationship between aerosols, air quality, weather and climate (Jones et al., 2001; Stier et al., 2005; Lohmann
et al., 2007; Benedetti et al., 2009; Morcrette et al., 2009; Kazil et al., 2010; Wang et al., 2011; Zhang et al., 2012; Forkel et al., 2015; Chan and Chan, 2017; Chan, 2017). These studies mostly rely on model simulations. However, atmospheric processing of aerosols is quite complex, and their physical and chemical properties are highly variable in space and time. Thus, simplifications and assumptions are required with respect to the physics and chemistry of aerosols and the computations of their radiative properties.

The current state of the description of radiative properties of particles in numerical models is elaborated e.g. by Baklanov et al. (2014). The influence of the description of particle microphysics, including their mixing state, hygroscopic growth and shape, on simulated aerosol optical properties was discussed e.g. in detail by Curci et al. (2015). Numerical simulations of atmospheric composition require meteorological data and chemical emission inventories as input. Emission inventories of anthropogenic pollutants can be estimated through the 'bottom-up' or 'top-down' method. The former one relies on statistics
of local information, such as road graph, industry location, population density, and electricity consumption, together with appropriate emission factors. The latter one uses observations as input and disaggregated to different emission sectors by means of local statistical indicators (van der Gon et al., 2012). Due to the rapid changes of sources, emission inventories might be outdated in specific regions introducing large uncertainties in the model. Moreover, physical and chemical processes in the atmosphere are parameterized in models due to the intricacy of these processes, leading to additional uncertainties. As a
consequence, validation of model output against observational data becomes increasingly important.

The relevance of validation is documented by the establishment of international activities, e.g. the Air Quality Model Evaluation International Initiative (AQMEII, Rao et al. (2011)) when up to 20 groups provided model simulations of – among others – particulate matter. Common to almost all validation activities – except for in-situ-measurements of mass concentrations – is that they require the transformation of prognostic variables of the model, e.g. mass mixing ratios of a number of aerosol
components, to variables that can be measured. These are typically optical properties of the aerosols. Validation studies relying on in-situ measurements of near surface concentrations, e.g. from the Airbase and EMEP-networks, were conducted e.g. by Solazzo et al. (2012); Im et al. (2015). Measurements of aerosol optical depth (AOD) are mainly based on AERONET. Balzarini et al. (2015) compared AOD at 12 AERONET sites and in-situ measurements from ground-based networks to investigate the

performance of two chemical mechanisms of WRF-Chem (Grell et al., 2005; Fast et al., 2006). In the framework of AQMEII-2 modeled single scattering albedo, asymmetry parameter and AOD were also compared to AERONET data (Curci et al., 2015). AOD and Angström exponents from AERONET as well as AOD from spaceborne measurements were used for validation in the framework of the MACC-II reanalysis project (Cuevas et al., 2015). Investigations in how far range resolved measurements from active remote sensing systems can serve for model validation has been conducted in the last few years only. A combination of EARLINET and AERONET data for the Lidar-Radiometer Inversion Code (LIRIC, Chaikovsky et al. (2016)) was used by Binietoglou et al. (2015) for ten selected stations to investigate the accuracy of four dust transport models. Siomos et al. (2017) also used LIRIC and focused on the validation of aerosol mass concentration profiles for 22 cases over Thessaloniki, Greece. Mona et al. (2014) performed an intercomparison on the basis of extinction coefficient profiles from EARLINET data at Potenza, Italy, and the BSC-DREAM8b model covering 310 cases out of 12 years. These studies demonstrated impressively that the exploitation of range resolved measurements offers new perspectives for validation.

Quantitative range resolved aerosol parameters can be obtained from advanced lidar measurements. These lidar systems are however expensive in invest and maintenance, and continuous operation is only slowly developing. For these reasons, ceilometers might be a new option, though they are only simple single-wavelength low energy backscatter lidars. On the other hand they are eye-safe and can be operated continuously and fully automated, therefore making them suitable for setting up extended networks. In recent years, many synoptic observation stations have already been equipped with ceilometers and the number is still growing. Although ceilometers were originally designed for cloud height detection only, recent studies show that ceilometers are also able to measure aerosol profiles (Flentje et al., 2010; Wiegner and Geiß, 2012; Cazorla et al., 2017). If ceilometers are calibrated the primary output is the so-called attenuated backscatter $\beta^*$. Inversion of the signals provide the particle backscatter coefficient $\beta_p$ if the lidar ratio $S_p$ is known. As $\beta_p$ in the infrared spectral range is rather insensitive to errors of the lidar ratio $S_p$ this is typically not an issue. In contrast, the derivation of aerosol extinction coefficients $\alpha_p$ may be subject to large uncertainties due to an actually unknown lidar ratio. As a consequence, $\beta^*$ or $\beta_p$ are candidates for validating aerosol profiles derived from NWP-models. However, to our knowledge this approach has not yet been applied.

In this study, for the first time a comparison of aerosol profiles provided by the Copernicus Atmosphere Monitoring System (CAMS) with long term measurements of the ceilometer network measurements operated by the German weather service Weather Service (DWD) is presented. CAMS forecasts are quite relevant as it is often used to provide boundary conditions for regional chemistry transport models. In section 2, the ceilometer data and the aerosol description in the model are described. The concept used for the validation is discussed in section 3. The intercomparison discussed in section 4 comprises ceilometer measurements of one year (from 1 September 2015 to 31 August 2016) at 12 different stations in Germany and includes investigations of the importance of the numerical description of the hygroscopic growth and shape of the particles. Moreover, the agreement of the spatiotemporal distribution of dust particles during a Saharan dust event is discussed. A summary and suggestions for further studies conclude the paper.

## 2 Basis of the intercomparison

The comparison of 'aerosol profiles' derived from weather forecast models and retrieved from ceilometer measurements suffers from the fact that models and measurements do not provide the same physical quantity. In this section the output of the IFS and the ceilometers is described. This constitutes the basis for the determination of a common quantity for the intercomparison.

### 2.1 ECMWF-IFS: aerosol description

The European Centre for Medium-Range Weather Forecast Integrated Forecast System (ECMWF-IFS) is a comprehensive Earth-system model. An aerosol and chemistry module is coupled to the ECMWF-IFS by the Copernicus Atmosphere Monitoring Service (CAMS) to provide analysis and forecasts of atmospheric composition (Buizza et al., 1999; Rabier et al., 2000; Bechtold et al., 2008; Drusch et al., 2009; Dutra et al., 2013). In this study, daily forecast data are taken at 00:00 UTC resulting a forecast lead time of 0 - 21 hours. In the framework of Global and regional Earth-system Monitoring using Satellite and in-situ data (GEMS), concentrations of aerosol compounds were included as new prognostic variables into IFS (Morcrette et al., 2009; Benedetti et al., 2009). The parameterization of aerosol physics is mainly based on the concept of the LOA/LMD-Z model (Boucher et al., 2002; Reddy et al., 2005). Tropospheric aerosols are introduced in the model including two natural types, sea salt and dust, and three other types with significant anthropogenic contribution, i.e., sulfate, organic matter and black carbon. Stratospheric and volcanic aerosols are not considered in the present version.

The emission of sea salt and dust is controlled by the wind speed at a height of 10 m. Following the findings of Engelstaedter and Washington (2007), it was suggested by Morcrette et al. (2008) to also consider the gustiness of the wind. The sources for the anthropogenic aerosols are taken from external emission inventories, i.e., the Emission Database for Global Atmospheric Research (EDGAR, 2013), the Global Fire Emission Database (GFED, van der Werf et al. (2010)) and the Speciated Particulate Emission Wizard (SPEW) were used in the simulation. A detailed description of the sources of aerosols can be found in Dentener et al. (2006).

The above mentioned five aerosol types are further subdivided: natural aerosols are categorized into three different size bins each, whereas carbonaceous aerosols are differentiated into hydrophobic and hydrophilic particles. Sulfur is presented in the model in two forms, sulfur dioxide ($SO_2$) and sulfate ($SO_4$), the former one was assumed in gas phase while the latter is assumed in particulate phase. In total, the mass mixing ratios $m$ of 11 different aerosol types (see Table 1) are introduced as prognostic variables in the model.

Mass mixing ratio of these 11 types of aerosols are provided with a temporal resolution of 3 hours. The horizontal resolution of the original CAMS model output used in this study is approximately $0.7° \times 0.7°$. The data is then transformed to one of the following spatial coordinate systems: Spherical Harmonics (SH), Gaussian Grid (GG) or Latitude/Longitude (LL). From this archive we retrieved the data on a regular Latitude/Longitude grid with $1° \times 1°$ resolution. The vertical dimension of the model is separated into 60 pressure-sigma levels. Optical properties of aerosol, e.g., extinction coefficient, aerosol optical depth and backscatter coefficient, are not included in the output of the model, but calculated offline from the model output of aerosol mass mixing ratio. A more detailed description of the treatment of the aerosols can be found in Morcrette et al. (2009).

## 2.2 Aerosol microphysical properties

To determine the interaction between aerosols and radiation, the optical properties of each type are calculated for the short-wave and long-wave spectral range. In this context also the change of the optical properties with relative humidity is considered. In this study, we adapted the aerosol microphysical properties assumed in the previous study (Reddy et al., 2005), as the resulting aerosol optical depths were reported to agree well with observations (Morcrette et al., 2009). A brief description of the aerosol micropyhsical properties relevant for this study is presented in the following.

The particle size distribution is assumed to be lognormal with three parameters: $\sigma_{g,i}$ as the 'geometric standard deviation', i.e., the width of the distribution, $r_{0,i}$ as the modal radius, and $N_i$ as the total number concentration of particles of mode $i$. Thus, the size distribution (with $r$ as the particle's radius) consisting of $k$ modes is described by Eq. 1

$$N(r) = \sum_{i=1}^{k} \frac{N_i}{\sqrt{2\pi} \cdot \ln \sigma_{gi} \cdot r} \cdot \exp\left\{-\left(\frac{\ln r - \ln r_{0i}}{\sqrt{2} \cdot \ln \sigma_{gi}}\right)^2\right\} \tag{1}$$

with normally $k \leq 3$.

All aerosol types except sea salt are assumed to have a mono-modal lognormal distribution ($k=1$). Only for sea salt, a bi-modal lognormal distribution is assumed ($k=2$). The parameters $\sigma_g$ and $r_0$ characterizing each aerosol type are listed in Table 1. They are based on Reddy et al. (2005) and valid for dry particles.

For the sulfate, organic matter and black carbon aerosol types $\sigma_g = 2.0$ is selected and the modal radii $r_0$ are 0.0355, 0.0355 and 0.0118 $\mu$m, respectively (Boucher and Anderson, 1995; Köpke et al., 1997). The microphysical properties of hydrophilic and hydrophobic carbonaceous aerosols are assumed to be the same.

Dust aerosols are also described by a mono-modal lognormal size distribution with $r_0 = 0.29\,\mu$m and $\sigma_g = 2.0$ (Guelle et al., 2000), but split into three size bins. The limits are 0.03 - 0.55 $\mu$m (fine mode), 0.55 - 0.9 $\mu$m (accumulation mode) and 0.9 - 20.0 $\mu$m (coarse mode), respectively. These boundaries are chosen so that approximately 10, 20 and 70 % of the total mass of the aerosols are in each of the size bins (Morcrette et al., 2009).

Sea salt aerosols as the second class of natural aerosols are also represented by three size bins. For dry sea salt aerosol, their limits are slightly different and set to 0.015 $\mu$m, 0.251 $\mu$m, 2.515 $\mu$m and 10.060 $\mu$m. In contrast to the dust aerosols, a bimodal lognormal with $r_0 = 0.1002\,\mu$m and 1.002 $\mu$m and $\sigma_g = 1.9$ and 2.0 (O'Dowd et al., 1997) is assumed. The number concentrations $N_1$ and $N_2$ of the first and second mode are 70 and 3 cm$^{-1}$, respectively.

The refractive index of sea salt is assumed to be wavelength independent (Shettle and Fenn, 1979). For all other aerosol types a wavelength dependence is assumed and tabulated for 44 wavelengths between $\lambda$=0.28 $\mu$m and 4.0 $\mu$m, with values taken from Boucher and Anderson (1995); Köpke et al. (1997) and Dubovik et al. (2002). The aerosol microphysical and optical properties at 1064 nm, the wavelength of the ceilometers of the DWD-network, are listed in Table 1. Other relevant wavelengths for ceilometer and lidar applications are listed in Table A1 in the appendix.

In case of hygroscopic growth of particles, their microphysical properties change. Typically this effect is parameterized by an increasing modal radius and limits of integration over the size distribution, whereas the width of the distribution $\sigma_g$ is

**Table 1.** Microphysical properties and selected optical properties of dry aerosols as used in the model.

| Aerosol Type | Wavelength ($\lambda$, nm) | Density ($\varrho_p$, g/cm$^3$) | Modal Radius (r$_0$, $\mu$m) | Geometric Standard Deviation ($\sigma_g$) | Refractive Index ($n$) | Single Scattering Albedo ($\omega_0$) | Specific Extinction Cross Section ($\sigma_e^*$, m$^2$/g) | Lidar Ratio ($S_p$, sr) |
|---|---|---|---|---|---|---|---|---|
| Sea Salt (bin 1)[a] | 1064 | 2.160 | 0.1002,1.0020[c] | 1.9,2.0[c] | 1.5156-0.0002i | 1.00 | 0.55 | 21.7 |
| Sea Salt (bin 2)[a] | 1064 | 2.160 | 0.1002,1.0020[c] | 1.9,2.0[c] | 1.5156-0.0002i | 1.00 | 0.62 | 10.0 |
| Sea Salt (bin 3)[a] | 1064 | 2.160 | 0.1002,1.0020[c] | 1.9,2.0[c] | 1.5156-0.0002i | 0.99 | 0.18 | 18.2 |
| Dust (bin 1)[b] | 1064 | 2.610 | 0.2900 | 2.0 | 1.4800-0.0006i | 1.00 | 1.50 | 78.6 |
| Dust (bin 2)[b] | 1064 | 2.610 | 0.2900 | 2.0 | 1.4800-0.0006i | 1.00 | 1.61 | 48.6 |
| Dust (bin 3)[b] | 1064 | 2.610 | 0.2900 | 2.0 | 1.4800-0.0006i | 0.99 | 0.44 | 13.4 |
| Organic Matter | 1064 | 1.769 | 0.0355 | 2.0 | 1.5068-0.0000i | 1.00 | 0.77 | 34.2 |
| Black Carbon | 1064 | 1.000 | 0.0118 | 2.0 | 1.7500-0.4500i | 0.08 | 3.90 | 168.3 |
| Sulfate | 1064 | 1.769 | 0.0355 | 2.0 | 1.5068-0.0000i | 1.00 | 0.77 | 34.2 |

[a] Sea salt aerosols are represented in the model by three size bins with the bin limits set to 0.015-0.251 $\mu$m (bin 1), 0.251-2.515 $\mu$m (bin 2) and 2.515-10.060 $\mu$m (bin 3).
[b] Dust aerosols are represented in the model by three size bins with the bin limits are set to 0.03-0.55 $\mu$m (bin 1), 0.55-0.90 $\mu$m (bin 2) and 0.90-20.00 $\mu$m (bin 3).
[c] A bimodal lognormal size distribution is assumed for sea salt aerosols, with r$_0$=0.1002 $\mu$m and 1.002 $\mu$m and $\sigma_g$=1.9 and 2.0. The number concentrations $N_1$ and $N_2$ of the first and second mode are 70 and 3 cm$^{-1}$, respectively.
Note that density of hydrophilic aerosol changes with hygroscopic growth of particle.

assumed to remain unchanged. The latter approximation is certainly a simplification, but frequently used. Hygroscopic growth is considered for sulfate, hydrophilic organic matter and sea salt, see Fig. 1. It is parameterized by growth factors, defined as the ratio between the radius of the wet and dry particle ($r/r_{dry}$) and taken from the OPAC database (Hess et al., 1998). For sulfate and hydrophilic organic matter the same factors are used. Especially for a relative humidity above 70 % the growth is
strong whereas no growth is assumed when the relative humidity is below 30 %. The refractive index $n$ and density $\varrho$ of wet particles is taken from a look up table with mixing rules following Hess et al. (1998).

To reduce computational time, the optical properties of hygroscopic aerosols are pre-calculated for 12 discrete relative humidity levels (0, 10, 20, 30, 40, 50, 60, 70, 80, 85, 90 and 95 %) and stored in a look-up table. It is important to note that sea salt aerosols are emitted and transported as wet aerosols in the model with properties equivalent to 80 % relative humidity.
Subsequently, the model reported mass mixing ratios of sea salt are converted back to dry aerosols. This conversion is achieved by dividing the model reported mass mixing ratios by the mass growth factor at 80 % relative humidity. The hygroscopic growth effect is then applied to the dry sea salt aerosols to determine the actual optical properties.

## 2.3 The ceilometer network

In recent years, DWD has equipped a number of synoptic observation stations with Lufft (previously Jenoptik) ceilometers
(CHM15k) to establish a ceilometer network (www.dwd.de/ceilomap). By the end of 2016, 100 ceilometers are put into operation in Germany. The locations of the ceilometer sites are indicated in Fig. 2. The ceilometer network is still expanding in order to have a better spatial coverage. The ceilometers are eye-safe and fully-automated systems which allow unattended operation on a 24/7 basis (Wiegner et al., 2014). They are suitable for monitoring aerosol layers (e.g., volcanic ash, see Flentje et al., 2010), validation of meteorological and chemistry transport models (see e.g. Emeis et al., 2011), and are foreseen for
data assimilation (e.g., Wang et al., 2014; Geisinger et al., 2017).

The CHM15k-ceilometer is equipped with a diode-pumped Nd:YAG-laser emitting laser pulses at 1064 nm. The typical pulse energy of the laser is about 8 $\mu$J with a pulse repetition frequency of 5 - 7 kHz. Backscattered photons are collected by the telescope through a narrow band interference filter and measured by an avalanche photodiode running in photon counting

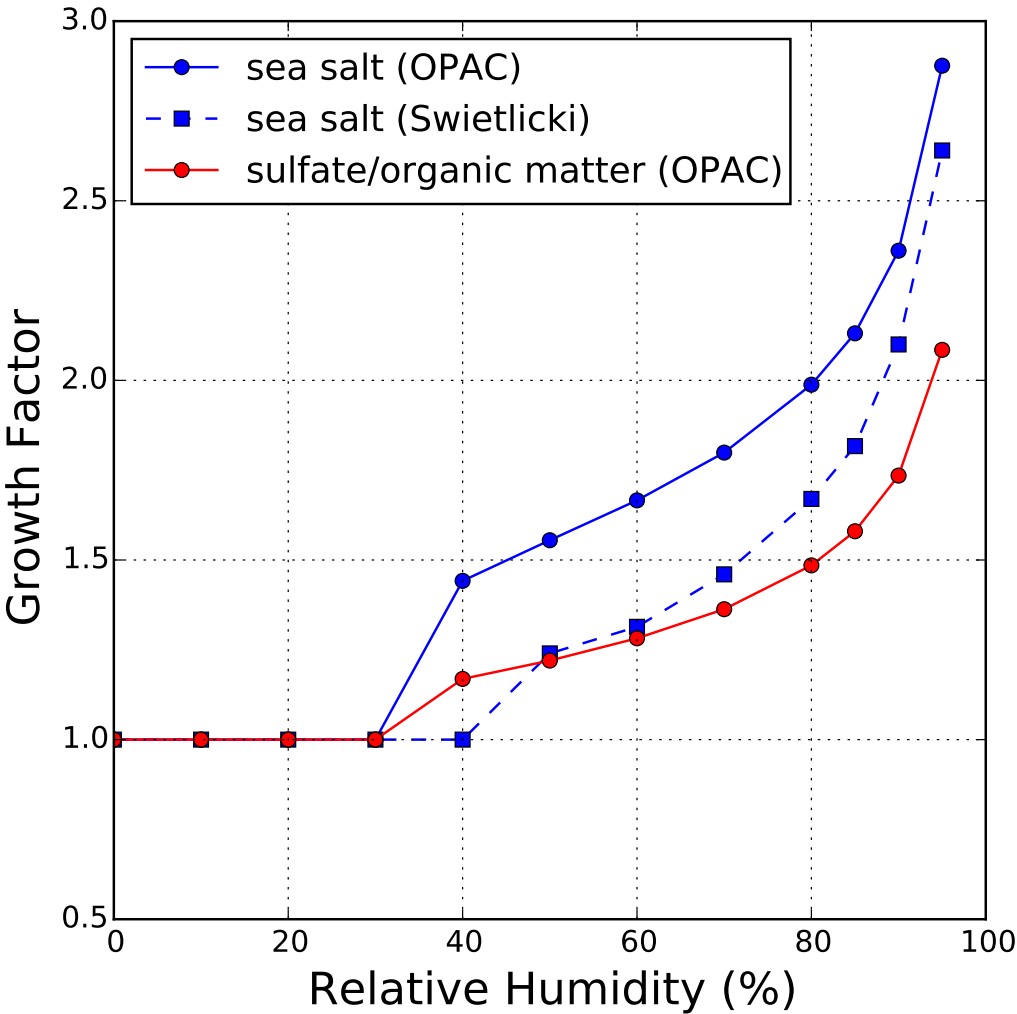

**Figure 1.** Hygroscopic growth factors of particle radius of sea salt, sulfate and hydrophilic organic matter aerosols as a function of relative humidity. Sulfate and hydrophilic organic matter share the same growth factor in the model (red curve). Growth factors of sea salt obtained from Swietlicki et al. (2008) are also shown for reference.

mode. The received backscatter signals are stored in 1024 range bins with a resolution of 15 m, the temporal resolution is set to 15 s. The signals are corrected for incomplete overlap by a correction function provided by the manufacturer. As ceilometers are single-wavelength backscatter lidars the received signals follow the well known lidar equation. Calibration is required to retrieve quantitative results.

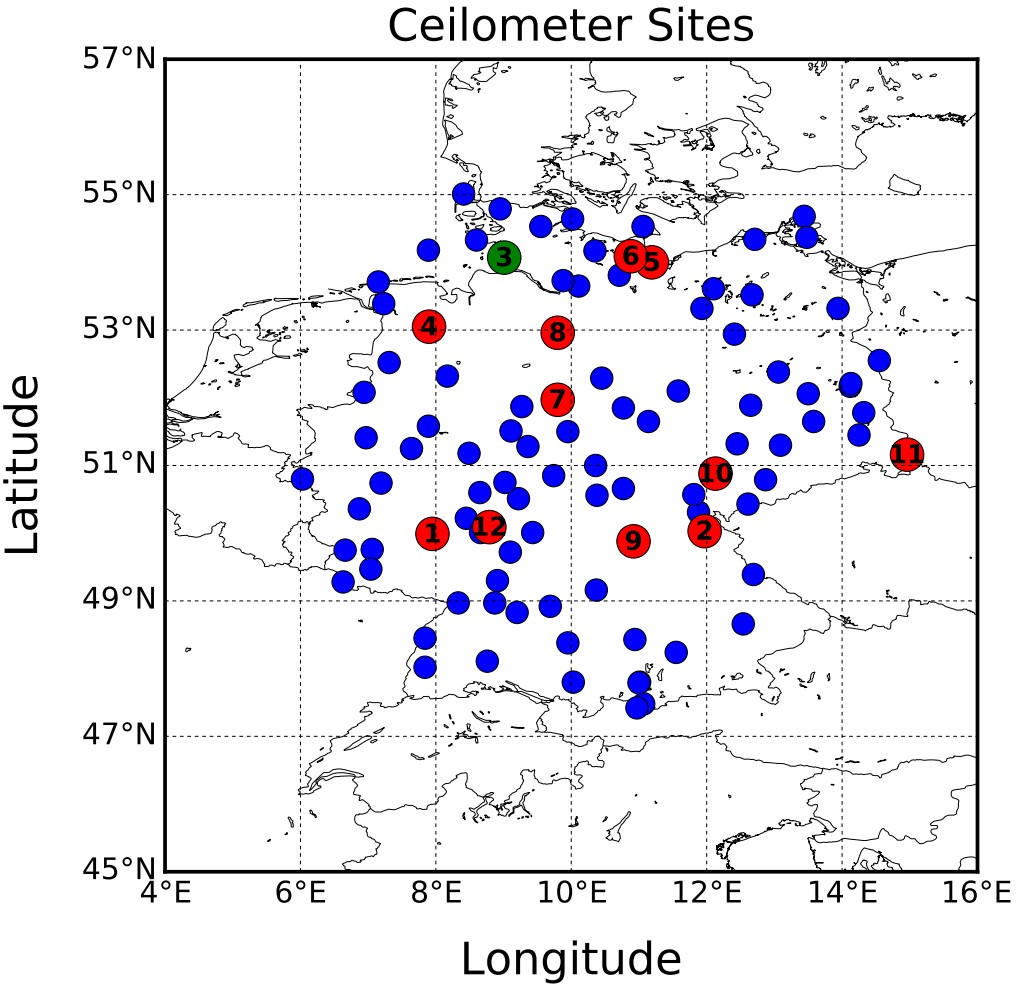

**Figure 2.** Location of the German weather service ceilometer sites as on end of 2017. The red spots indicate the ceilometer sites within 20 km of IFS model grid point while the blue markers represent the rest of the network. Note that some of the sites are not in operation during the period of study and therefore not included in this study. Elpersbüttel (see section 4) is indicated in green. More detailed information of the ceilometer sites can be found in Table 2.

For the intercomparison of ceilometer measurements and modeled aerosol profiles we only consider sites within 20 km from a model grid point. This criterion results in a selection of 12 stations. Their location (latitude, longitude, altitude) together with their distance from the nearest IFS-grid point are summarized in Table 2.

## 3 Concept of intercomparison

As mentioned above, profiles of mass mixing ratios cannot directly be compared to 'ceilometer profiles'. The latter can be expressed as particle backscatter coefficient $\beta_p(z)$ or as attenuated backscatter $\beta^*(z)$

$$\beta^*(z) = \beta(z)\,\exp\left\{-2\int\limits_0^z \alpha(z')\mathrm{d}z'\right\} \tag{2}$$

with $z$ being the height, $\beta$ and $\alpha$ the backscatter and the extinction coefficient, respectively. From the model results $\beta^*(z)$ and $\beta_p(z)$ can be calculated straight forward and the computational effort is comparable. Retrieval of both $\beta^*(z)$ and $\beta_p(z)$ from ceilometer measurements require the calibration of the ceilometer, i.e. the determination of the lidar constant $C_L$ (also known as calibration factor). The derivation of $\beta_p(z)$ requires furthermore an inversion of the signals (e.g. Klett, 1981; Fernald, 1984) relying on the assumption of a particle lidar ratio $S_p$, which depends on the aerosol composition. Consequently, additional uncertainties are introduced. It can be expected that the relative error of $\beta_p$ is as good as approximately 15 % for specific Lufft ceilometers (Wiegner and Geiß, 2012) but can also exceed 30%. Note that water vapor absorption must be taken into account for ceilometers operating near 910 nm, otherwise an additional uncertainty depending on the water vapor content and the spectrum of the emitted laser radiation is introduced (Wiegner and Gasteiger, 2015). Fortunately, this does not apply for the ceilometers of the DWD which measure at 1064 nm. However, this effect may be relevant for other ceilometer networks.

For these reasons, and because weather services are in favor of the attenuated backscatter for intercomparisons, we chose $\beta^*(z)$ as the common quantity in this study. In this section, the procedures to derive attenuated backscatter from model simulations and ceilometer measurements are presented in detail.

### 3.1 Attenuated backscatter from model output

The model outputs consist of the mass mixing ratios $m_{p,i}$ of the 11 aerosol types and no optical property of aerosol is provided. Therefore, we have to convert the model output to attenuated backscatter to compare to ceilometer measurements. In a first step the mass mixing ratios of each aerosol type are converted to mass concentrations $c_{p,i}$ by multiplying with the air density $\varrho_{\mathrm{air}}$ as shown in Eq. (3), with the air density calculated from the temperature and pressure profiles of the IFS-model.

$$c_{p,i}(z) = \varrho_{\mathrm{air}}(z)\,m_{p,i}(z) \qquad \text{for i=1,2,...,11} \tag{3}$$

The particle extinction coefficient $\alpha_{p,i}$ of each aerosol type $i$ is calculated using fundamental relations of scattering theory as shown in Eq. (4).

$$\alpha_{p,i}(z) = \pi \int\limits_{r_1}^{r_2} r^2\, Q_{\mathrm{ext,i}}(z)\, \frac{dN_i(r)}{dr}\, dr \tag{4}$$

where $Q_{\text{ext,i}}$ is the extinction efficiency, and $r_1$ and $r_2$ the lower and upper limits of the size bin. The particle backscatter coefficient is defined in a similar way:

$$\beta_{p,i}(z) = \pi \int\limits_{r_1}^{r_2} r^2 \, Q_{\text{bsc,i}}(z) \frac{dN_i(r)}{dr} \, dr \tag{5}$$

with $Q_{\text{bsc,i}}$ being the scattering efficiency multiplied with the phase function at $180°$. For convenience it is common to use
the lidar ratio $S_{p,i}$

$$S_{p,i}(z) = \frac{\alpha_{p,i}(z)}{\beta_{p,i}(z)} \tag{6}$$

to calculate particle backscatter coefficients from extinction coefficients.

The extinction efficiencies and lidar ratios of each aerosol type are calculated applying the size distribution $dN(r)/dr$ and the refractive index $n$ of the particles, and by means of an appropriate scattering theory: for spherical particles the Mie theory
is applied, for nonspherical particles a suite of approaches is available with the T-matrix (Mishchenko and Travis, 1998) being the most frequently used option. As reference, we use the Lorenz-Mie scattering algorithm (Mishchenko et al., 1999) even for nonspherical aerosol types as dust, but include a detailed discussion of the influence of particle shape on lidar related optical properties in section 4.1.2. In order to retrieve the optical properties of the 11 aerosol types integration was performed according to the given size bins, otherwise the upper limit was set to $r = 20\,\mu$m. In case of hygroscopic growth of particles,
their physical size, refractive index and density change according to the look up table mentioned above.

The conversion from the mass concentration to the extinction coefficient can now readily be achieved by using the mass extinction coefficient $\eta_{\alpha,i}$ (given e.g. in m$^2$/g)

$$\eta_{\alpha,i} = \frac{\alpha_{p,i}}{c_{p,i}} = \frac{3 \int_{r_1}^{r_2} r^2 \, Q_{\text{ext,i}} \, (dN_i(r)/dr) \, dr}{4 \, \varrho_p \int_{r_1}^{r_2} r^3 \, (dN_i(r)/dr) \, dr} \tag{7}$$

in the radius interval of the corresponding size bin from $r_1$ to $r_2$. Finally, the extinction and backscatter coefficients of each
aerosol type are determined – with consideration of Eq. (6) – according to

$$\begin{aligned} \alpha_{p,i} &= c_{p,i} \, \eta_{\alpha,i} \\ \beta_{p,i} &= c_{p,i} \left( \frac{\eta_{\alpha,i}}{S_{p,i}} \right) = c_{p,i} \, \eta_{\beta,i} \end{aligned} \tag{8}$$

Here, $\eta_{\beta,i}$ is the factor converting mass concentration to backscatter coefficient (of aerosol type $i$). The contribution of the air molecules is determined from the Rayleigh theory. We use the following approximation for the extinction coefficient $\alpha_m$
(in km$^{-1}$)

$$\alpha_m(z,\lambda) = 8.022 \cdot 10^{-4} \varrho_{\text{air}}(z) \lambda^{-4.08}$$

with the air density given in kg/m$^3$ and the wavelength $\lambda$ in $\mu$m. The molecular lidar ratio $S_m$ is known to be

$$S_m = \frac{\alpha_m}{\beta_m} \approx \frac{8\pi}{3}$$

Finally, we have to take all contributions into account, i.e., the (total) extinction coefficient $\alpha$ is determined according to

$$\alpha = \alpha_m + \sum_{i=1}^{11} \alpha_{p,i} + \alpha_w \tag{9}$$

and the (total) backscatter coefficient is

$$\beta = \beta_m + \sum_{i=1}^{11} \beta_{p,i} \tag{10}$$

Ultimately, the attenuated backscatter $\beta^*(z)$ can be calculated as described in Eq. (2). Note, that the effective water vapor absorption coefficient $\alpha_w$ must only be considered in Eq. (9) if model results shall be compared to ceilometers operating in the spectral range around 910 nm (Wiegner and Gasteiger, 2015). This is e.g. the case if Vaisala-ceilometers were applied.

To increase the efficiency of the computations, $\eta_{\alpha,i}$ and $S_{p,i}$ are pre-calculated. An overview of aerosols in dry conditions for the ceilometer wavelength (1064 nm) is given in Table 1. The wavelengths corresponding to Nd:YAG-lasers used for aerosol remote sensing (355 nm, 532 nm and 1064 nm), the widely used Vaisala ceilometers (910 nm), and the 'typical wavelength' for radiative transfer calculations in the shortwave spectral range (550 nm) are also shown in Table A1 in the appendix. Note, that the lidar ratios of some aerosol types differ from values published by, e.g., Groß et al. (2015) because of the limits of the particle size bins.

## 3.2 Attenuated backscatter from ceilometers

Attenuated backscatter $\beta^*$ can be derived from the background corrected ceilometer signals $P$ if the system has been calibrated, i.e., if $C_L$ is known.

$$\beta^*(z) = \frac{Pz^2}{C_L} \tag{11}$$

It should be emphasized that $C_L$ can vary with time (e.g. caused by aging of components, or temperature drifts), thus calibration should be performed on a regular basis whenever weather conditions permit.

**Table 2.** Ceilometer sites within a distance of 20 km to the nearest IFS model grid point, altitude is given in meters above mean sea level, the distance to the nearest model grid point (in km) is given in the 5. column.

| No. | Site | Latitude (°N) | Longitude (°E) | Altitude (m) | Distance (km) |
|---|---|---|---|---|---|
| 1 | Geisenheim | 49.9866 | 7.9551 | 110 | 3.8 |
| 2 | Wunsiedel | 50.0316 | 11.9745 | 622 | 4.0 |
| 3 | Elpersbüttel | 54.0692 | 9.0105 | 3 | 7.8 |
| 4 | Friesoythe | 53.0500 | 7.9000 | 6 | 8.7 |
| 5 | Boltenhagen | 54.0027 | 11.1909 | 15 | 12.5 |
| 6 | Pelzerhaken | 54.0893 | 10.8773 | 1 | 12.7 |
| 7 | Alfeld | 51.9644 | 9.8072 | 144 | 14.1 |
| 8 | Soltau | 52.9605 | 9.7930 | 76 | 14.2 |
| 9 | Bamberg | 49.8743 | 10.9206 | 240 | 14.5 |
| 10 | Gera | 50.8813 | 12.1289 | 311 | 16.2 |
| 11 | Görlitz | 51.1633 | 14.9531 | 240 | 18.0 |
| 12 | Offenbach | 50.0894 | 8.7864 | 121 | 18.1 |

The calibration of the ceilometers of the network is performed routinely by the DWD in a fully automated procedure. It is based on the TOPROF/E-Profile Rayleigh calibration routine provide by MeteoSwiss. The calibration relies on the Rayleigh method (Barrett and Ben-Dov, 1967). This is feasible under clear sky conditions and stable aerosol distributions, thus, the applicability depends on the measurement site. In order to avoid adverse influences caused by background sun light, only night time data are used for the calibration. The calibration is based on data averaged over 1 - 3 hours, and only one period is selected per night. Meteorological data used for the Rayleigh calibration are taken from the joint product of the National Centers for Environmental Prediction (NCEP) and the National Center for Atmospheric Research (NCAR) reanalysis data (Kalnay et al., 1996). The derived $C_L$ are first cleaned for outliers and then smoothed with a 30 days running mean. Calibration constants outside 1.5 times of the 25 to 75 percentile range of a 30 days-period are considered as outliers. The smoothed $C_L$ are finally interpolated to hourly values to be used in Eq. 11. The typical uncertainty of an individual calibration is 15 - 20 %, while the actual error is smaller due to the temporal smoothing. The accuracy of the retrieved $\beta^*$ linearly depends on the accuracy of the $C_L$. The monthly variation of $C_L$ is usually less than 5 % and the annual variation is 10 - 15 %. Then, attenuated backscatter $\beta^*$ profiles are derived in steps of three hours, by averaging cloud free data within 30 minutes each before and after the corresponding model time. Longer averages are desirable in view of a better signal to noise ratio but are critical during day time if the aerosol distribution is rapidly changing in time. In cases of rain, fog, snowfall and low level clouds (below 2 km), the data are excluded from the evaluation. The data quality flag 'sky-condition-index' from the proprietary software of the ceilometer labels corresponding measurements and cases of reduced window transmission due to droplets on the window. The

altitude of the cloud bottom is determined by a complex algorithm based on signal slopes and thresholds; the details are not disclosed to the user.

## 4 Results and discussions

There are several options to discuss the agreement of $\beta^*$-profiles from model calculations and ceilometer measurements: criteria include the comparison of absolute values of $\beta^*$, the general 'shape' of the profiles, the vertical extent of the mixing layer and elevated layers, the vertical structure of the aerosol distribution within the mixing layer, and more. A general philosophy on a ranking of different criteria has not yet been developed, e.g., there is no common agreement how to rate profiles when the modeled altitude of an elevated layer is consistent with measurements but the absolute values of $\beta^*$ are different. The reason is that the attenuated backscatter of e.g. an elevated Saharan dust or volcanic ash layers may disagree even in the case of the same $\beta_p$ because $\beta^*$ does not only depend on the aerosol properties of that layer, but is also influenced by the extinction below that layer. In order to minimize this influence and to consider that part of the atmosphere where most of the aerosols typically reside, we focus in this paper on $\beta^*$ of the lowermost part of the troposphere excluding the range of incomplete overlap. All ceilometer data have undergone an individual overlap correction provided by the manufacturer that makes it possible to use profiles for aerosol remote sensing from above approximately 200 m. In the following, we compare $\beta^*$ averaged from the typical height of a 'reliable overlap correction' (set to 200 m for all instruments) to 1 km above ground, henceforward referred to as 'near surface average' $\overline{\beta^*_{ns}}$. An additional approach of comparison is discussed in Sect. 4.2.

Our investigation is based on measurements from 1 September 2015 until 31 August 2016. Attenuated backscatter profiles are derived from the model results for every 3 hours following the procedure outlined in section 3.1. Ceilometer data are averaged over 1 hour around the model time and profiles with low level clouds and precipitation are excluded from the analysis. As a consequence, averages consider 240 ceilometer profiles at maximum.

### 4.1 Comparison of near surface attenuated backscatter

For an overview the complete time series of the model simulation and ceilometer observation of the near surface attenuated backscatter $\overline{\beta^*_{ns}}$ over Elpersbüttel is shown in Fig. 3. This site has been chosen as it is one of the closest to the corresponding model grid point (only 7.8 km south west of the ceilometer site, see Table 2) and the orography around the measurement site is quite flat. For Elpersbüttel we found 1305 cases out of 2920 ($365 \times 8$) when intercomparisons could take place. The number of cases varies in a range from 900 (Wunsiedel) to 1763 (Boltenhagen). Results show that the model and ceilometer data both show a similar temporal development with larger $\overline{\beta^*_{ns}}$ during winter and spring. Note, that due to cloudy weather during winter the number of useful ceilometer measurements is reduced compared to summer. In cases of low aerosol load there is a general agreement of both data sets. However, when episodes of large values of $\overline{\beta^*_{ns}}$ are modeled they typically exceed the observed ones by a factor of two or more. This is e.g. the case in December 2015, beginning of February 2016 and April 2016, more detailed investigations are presented in Section 4.1.2 and Section 4.2. The reasons for this overestimate must remain speculative - maybe it is due to erroneous assumptions of the aerosol emission or meteorological data. On the other

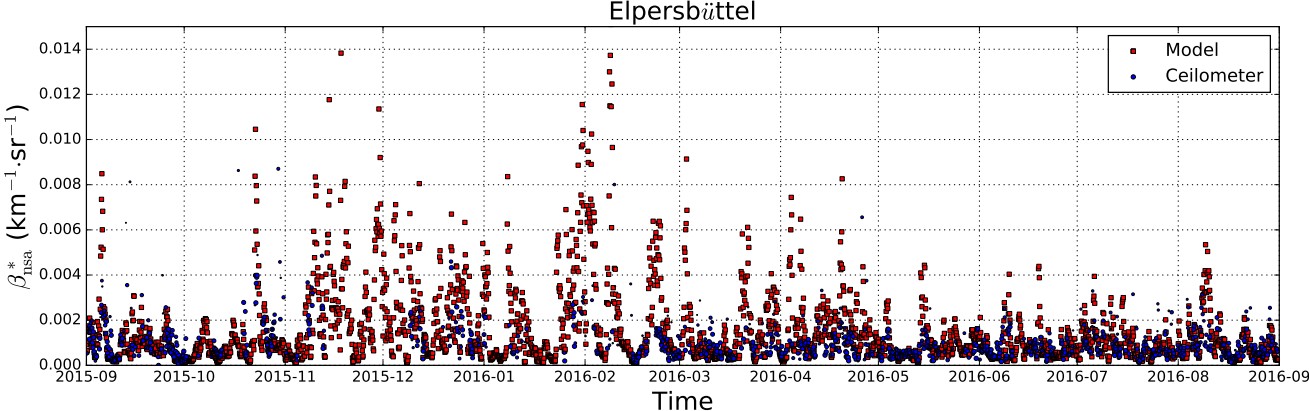

**Figure 3.** Time series (from 1 September 2015 to 31 August 2016) of the model simulation and ceilometer observation of near surface attenuated backscatter $\overline{\beta_{ns}^*}$ in km$^{-1}$ sr$^{-1}$ over Elpersbüttel with $\overline{\beta_{ns}^*}$ being averaged from 200 m ("full overlap height") up to 1 km above ground.

hand the annual mean derived from the model $\overline{\beta_{ns}^*} = 1.35 \times 10^{-3}$ km$^{-1}$ sr$^{-1}$ agrees very well with the corresponding value of $\overline{\beta_{ns}^*} = 1.31 \times 10^{-3}$ km$^{-1}$ sr$^{-1}$ from the ceilometer observations at Elpersbüttel. Table 3 summarized the annual mean $\overline{\beta_{ns}^*}$ of both ceilometer measurements and model simulations from all 12 measurement sites. For most of the sites, the model obtains higher values than the ceilometer measurements by $\sim$20 % on annual average. This is still a considerably good agreement.

Few exceptions are Bamberg, Boltenhagen and Gera where the model predicts much lower values than observed. The largest difference of a factor of 1.8 is found for Gera with measurement and model mean values of $\overline{\beta_{ns}^*} = 1.85 \times 10^{-3}$ km$^{-1}$ sr$^{-1}$ and $\overline{\beta_{ns}^*} = 1.00 \times 10^{-3}$ km$^{-1}$ sr$^{-1}$, respectively. These stations show larger impacts from the local emissions as they are situated close to the cities. The discrepancy between ceilometer observations and model predictions over these three sites is mainly due to the differences in the spatial coverage. As the model resolution is rather coarse ($1° \times 1°$), the model is underestimating the

aerosol concentrations over cities due to the averaging over large grid cell.

For a more detailed analysis we have calculated the differences $\Delta$ between modeled and ceilometer derived $\overline{\beta_{ns}^*}$ with

$$\Delta = \overline{\beta_{ns}^*}(\text{mod}) - \overline{\beta_{ns}^*}(\text{obs}) \qquad (12)$$

for Elpersbüttel (see Fig. 4a). The size of the markers is proportional to the number of ceilometer measurements (up to 240) available for each individual intercomparison. The standard deviation $\sigma$ of the difference is $\sigma = 1.89 \times 10^{-3}$ km$^{-1}$ sr$^{-1}$, i.e.

quite large compared to the model mean value of $\overline{\beta_{ns}^*} = 1.35 \times 10^{-3}$ km$^{-1}$ sr$^{-1}$. Data points with $|\Delta| > 3\sigma$ are considered as outliers (marked in red in Fig. 4a) and filtered out in the subsequent analysis. The remaining data are then used to recalculate the standard deviation ($\sigma = 1.20 \times 10^{-3}$ km$^{-1}$ sr$^{-1}$), shown as a horizontal line in Fig. 4a.

**Table 3.** Summary of the annual mean $\overline{\beta^*_{ns}}$ of both ceilometer measurements and model simulations from all 12 measurement sites. The standard deviation $\sigma$ of the differences is also indicated.

| No. | Site | annual average ceilometer $\overline{\beta^*_{ns}}$ ($\times 10^{-3}\,\mathrm{km^{-1}\,sr^{-1}}$) | annual average model $\overline{\beta^*_{ns}}$ ($\times 10^{-3}\,\mathrm{km^{-1}\,sr^{-1}}$) | standard deviation of difference $\sigma$ ($\times 10^{-3}\,\mathrm{km^{-1}\,sr^{-1}}$) |
|---|---|---|---|---|
| 1 | Geisenheim | 0.91 | 1.12 | 1.25 |
| 2 | Wunsiedel | 1.05 | 0.82 | 0.90 |
| 3 | Elpersbüttel | 1.31 | 1.35 | 1.20 |
| 4 | Friesoythe | 1.31 | 1.20 | 1.58 |
| 5 | Boltenhagen | 1.87 | 1.28 | 1.46 |
| 6 | Pelzerhaken | 1.13 | 1.24 | 1.13 |
| 7 | Alfeld | 1.07 | 1.13 | 1.31 |
| 8 | Soltau | 1.17 | 1.14 | 1.31 |
| 9 | Bamberg | 1.39 | 0.94 | 1.88 |
| 10 | Gera | 1.85 | 1.00 | 1.97 |
| 11 | Görlitz | 0.94 | 0.82 | 0.71 |
| 12 | Offenbach | 0.80 | 0.90 | 0.78 |

To better understand possible reasons for these differences we have looked into the contribution of different aerosol types. Their relative contributions to $\overline{\beta^*_{ns}}$ as calculated from the model for Elpersbüttel reveal that sea salt is by far the dominating contributor with 61 % (annual mean). Sulfate contributes with 29 % to the near surface attenuated backscatter, while organic matter (4 %), dust (3 %) and black carbon (2 %) only show minor contributions. We have re-calculated these contributions

separately for two classes: cases of 'good' agreement ($|\Delta| < \sigma$) are shown in Fig. 4b, whereas cases of 'bad' agreement ($|\Delta| > \sigma$) are shown in Fig. 4c. Each aerosol type is color coded as indicated in the legend.

From Fig. 4b it is immediately visible that for the good agreement sea salt is again the dominating aerosol type: its contribution ranges between 32 % (May 2016) and 85 % (December 2015) with an annual average of 51 %. The second important contributor are sulfate aerosols (32 % on average) whereas all other types are in the range of a few percent each. Thus, cases of

good agreement coincide with a sea salt contribution lower than the mean. Consequently, the contribution of sea salt is above the average when the differences between model and measurement are large ($|\Delta| > \sigma$). From Fig. 4c a mean relative contribution of sea salt of 74 % for the 'bad' agreement is found. This suggests that the ceilometer and model discrepancy increases with increasing sea salt contribution.

Scatter plots of the ceilometer and the model derived near surface attenuated backscatter for the 12 sites are shown in Fig. 5.

The color code represents the relative contribution of sea salt to $\overline{\beta^*_{ns}}$. For most sites red dots are predominant, indicating the high contribution of sea salt. This phenomenon has already been discussed in case of Elpersbüttel. When the sea salt contribution is rather low, the model typically shows lower $\overline{\beta^*_{ns}}$ than the ceilometer retrieval. This is probably due to local

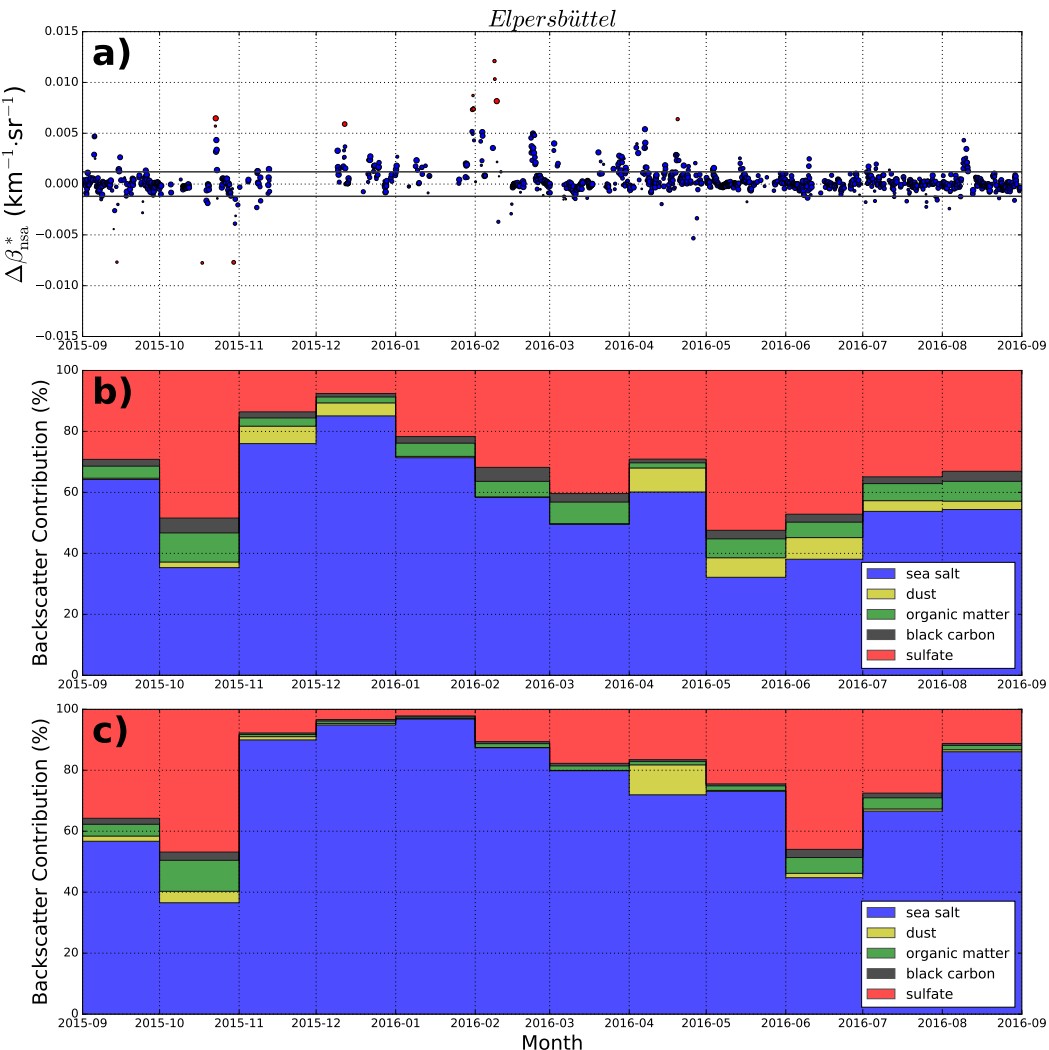

**Figure 4.** (a) Differences $\Delta$ of near surface attenuated backscatter $\overline{\beta_{ns}^*}$ for Elpersbüttel according to Eq. 12. The horizontal line indicates the standard deviation $\sigma$ of the differences. The size of the markers represents the number of ceilometer measurements available for each individual comparison. Differences $|\Delta| > 3\sigma$ are shown in red. (b) Contribution of different aerosol types to $\overline{\beta_{ns}^*}$ for cases with $|\Delta| < \sigma$. (c) same as (b) but cases with $|\Delta| > \sigma$.

emissions which are not well resolved by the model but captured by the ceilometer measurements. The total least squares regression line is based only on intercomparisons when the hourly averaged data contains at least 120 ceilometer profiles (30 minutes of measurements). The regression is virtually unchanged when the number of valid ceilometer profiles is used as a weight. The slope of the regression line is larger than 1 for all sites, indicating that the model in general results in larger $\overline{\beta_{ns}^*}$.

5   In particular this is true when the modeled contribution of sea salt is high, e.g. for Friesoythe, Geisenheim and Offenbach.

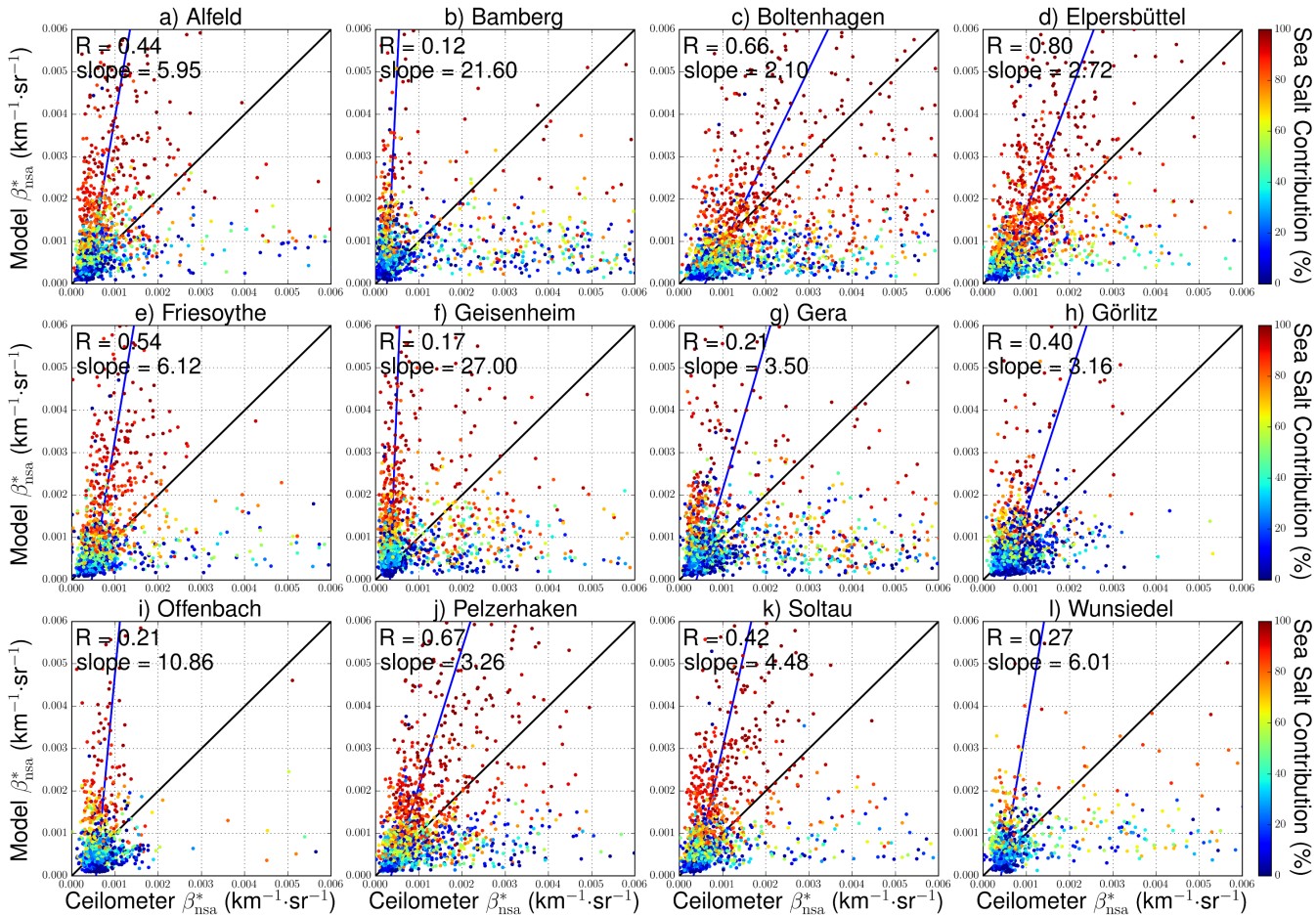

**Figure 5.** Scatter plots of ceilometer derived and modeled $\overline{\beta_{ns}^*}$ for the 12 sites listed in Table 2. The color code represents the relative contribution of sea salt to $\overline{\beta_{ns}^*}$. The blue curve indicates the total least squares regression line of the data points with at least 120 ceilometer profiles, while the black line represents the 1 to 1 reference.

Note, that the latter two stations are far from the coast so that the large sea salt contribution seems to be unrealistic. Pearson's correlation coefficient $R$ ranges between $R = 0.12$ for Bamberg and $R = 0.80$ for Elpersbüttel with no clear dependence on the distance between the model grid point and the ceilometer site.

Reasons for the disagreement can be manifold: One possibility is that the backscatter per unit mass of sea salt is too large in the model. As the optical properties of sea salt critically depend on the hygroscopic growth we have investigated to which extent this effect might explain the observed differences (see section 4.1.1). Another reason could be that the modeled sea salt concentration is generally overestimated, though the annual averaged contribution to the total aerosol optical depth (AOD) is ranging from 21 % (Görlitz) to 37 % (Elpersbüttel), which is in a reasonable range. One the other hand this is much less

than the contribution to $\overline{\beta^*_{ns}}$ demonstrating that sea salt is quite effectively backscattering, suggesting that it might partly be substituted by a less effective species to get a better agreement. A further discussion of this topic is however beyond the scope of this paper.

### 4.1.1 Influence of hygroscopic growth

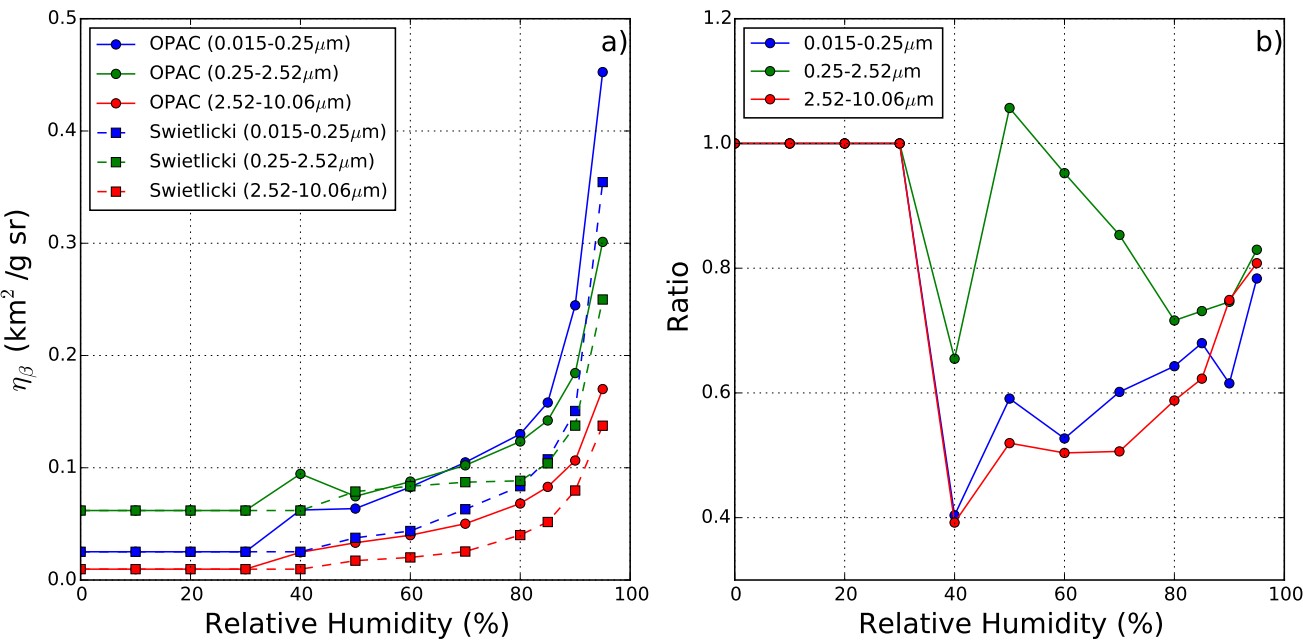

**Figure 6.** a) Mass mixing ratio to backscatter coefficient conversion factors $\eta_{\beta,i}$ of sea salt aerosol for the three small, medium and large size bin (in blue, green and red) at different ambient relative humidities. Hygroscopic growth factors of sea salt are taken from the model assumption (solid circle curve) and Swietlicki et al. (2008) (dashed square curve). The ratio of mass mixing ratio to backscatter coefficient conversion factors between the hygroscopic growth effects taken from Swietlicki et al. (2008) and OPAC databases are shown in b), ratios smaller than 1 indicate an reduction when using hygroscopic growth from Swietlicki et al. (2008).

5      Water uptake by particles has a significant impact on their optical properties as particles can change in size, chemical composition and refractive index depending on the ambient relative humidity. The assumptions made for their hygroscopic growth have a significant effect on the simulation of ceilometer measurements from the model output. For this reason, we examine the hygroscopic growth effect on the conversion factor $\eta_\beta$ for sea salt (see Eq. 8) as the dominating aerosol species (in terms of backscatter) according to the IFS output. We compare two approaches, being aware that more are existing (e.g., Chin et al.,

10    2002): the particle hygroscopic growth model implemented in the IFS AOD calculation (based on OPAC, Hess et al. (1998)) and a more recent approach (Swietlicki et al., 2008), see Fig. 1. The latter was reported to better match experimental data

(Zieger et al., 2013). Compared to OPAC it shows a less pronounced particle growth with relative humidity. The corresponding conversion factors $\eta_{\beta,ss}$ of sea salt are shown in Fig. 6a for comparison. Results referring to the three different size bins of the particle model are shown in blue, green and red, respectively. The ratios of the conversion factors from the two approximations ($\eta_{\beta,ss}^{(swie)}/\eta_{\beta,ss}^{(opac)}$) are shown in Fig. 6b. The conversion factors based on Swietlicki et al. (2008) are on average smaller than those based on the OPAC database. By comparing data calculated with the OPAC database, the alternative set of conversion factors $\eta_{\beta,ss}^{(swie)}$ on average reduce backscatter coefficients by a factor of 0.78. Taking into account that sea salt particles in general contribute more than 50 % to the attenuated backscatter over Germany, overestimating the conversion factor by 22 % on average would already contribute up to an error of more than 10 % of the total backscatter signal.

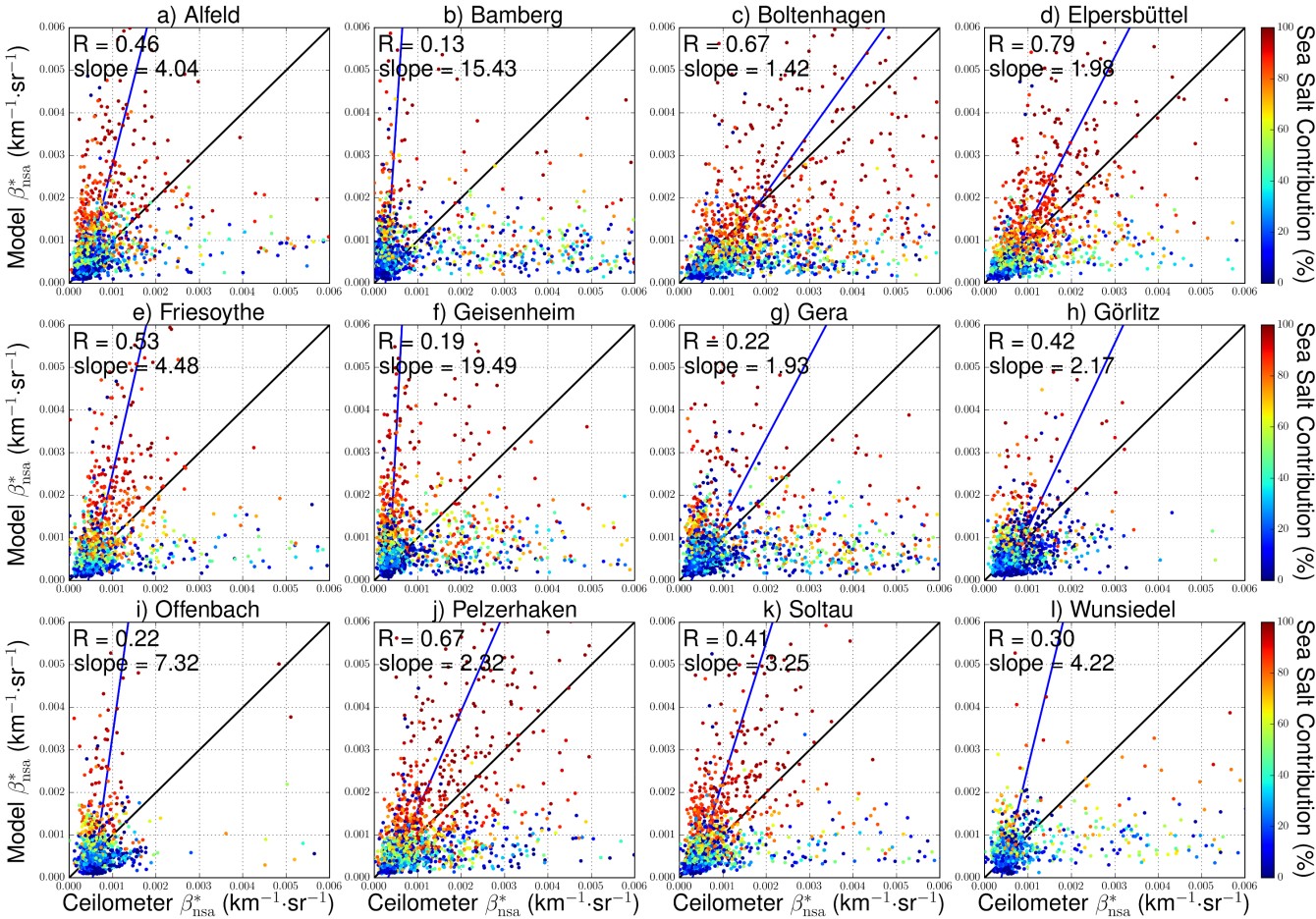

**Figure 7.** Scatter plots of the ceilometer and model surface attenuated backscatter signals for the 12 sites listed in Table 2. Model data are converted to attenuated backscatter signal based on hygroscopic growth factors introduced in Swietlicki et al. (2008). Color code represents the relative contribution of sea salt to backscatter signal. The blue curve indicates the total least squares regression line of data point with at least 120 ceilometer profiles, while the black curve represents the 1 to 1 reference.

In order to quantify the effect of a changed hygroscopic growth we recalculate $\beta^*$ from modeled mixing ratios by using the alternative set of conversion factors (Swietlicki et al., 2008) and compare it to ceilometer observations. Analogously to Fig. 5 scatter plots of the ceilometer derived and modeled $\overline{\beta_{ns}^*}$ for the 12 sites are shown in Fig. 7. Compared to the original model assumptions, modeled attenuated backscatter shows a slightly better agreement with the ceilometer measurements. Although the correlation coefficients between ceilometer and model $\overline{\beta_{ns}^*}$ are nearly unchanged, the slope of the regression lines is on average reduced by ∼30 % and agrees better with the 1 to 1 reference line. This effect is more obvious for sites dominated by sea salt aerosols in Northern Germany, e.g. Boltenhagen, Elpersbüttel, Pelzerhaken and Soltau. The result indicates that the updated hygroscopic growth function leads to a better agreement between model simulations and measurements. However, the model is still overestimating $\overline{\beta_{ns}^*}$, indicating that the assumption of a reduced hygroscopic growth alone cannot fully explain the mismatch between model and observations.

### 4.1.2 Influence of particle shape

**Table 4.** Comparison of selected optical properties at 1064 nm of mineral dust particles assuming spherical and nonspherical shapes. Spheroid particles with an aspect ratio distribution measured by Kandler et al. (2009) is assumed for nonspherical dust particles.

| species | spherical | | nonspherical | | difference |
|---|---|---|---|---|---|
| | specific extinction cross section ($\sigma_e^*$, m$^2$/g) | lidar ratio ($S_p$, sr) | specific extinction cross section ($\sigma_e^*$, m$^2$/g) | lidar ratio ($S_p$, sr) | in particle backscatter ($\Delta \beta_p$, %) |
| Dust (0.03 - 0.55 $\mu$m) | 1.496 | 78.6 | 1.449 | 89.0 | -14.4 |
| Dust (0.55 - 0.90 $\mu$m) | 1.611 | 48.6 | 1.602 | 69.4 | -30.3 |
| Dust (0.90 - 20.0 $\mu$m) | 0.445 | 13.4 | 0.495 | 26.6 | -44.0 |

Besides of the hygroscopic growth of hydrophilic aerosols, the shape of particles plays an important role for lidar related optical properties of particles. Mineral dust particles are typically nonspherical, however, they are often - e.g. in the IFS model - considered as spherical particles in order to simplify the computation. To quantify the influence of the shape, we compared modeled $\beta_p$ and $\beta^*$ using either the spherical or the nonspherical assumption. In case of nonspherical mineral dust particles, spheroids with an aspect ratio distribution measured by Kandler et al. (2009); Wiegner et al. (2011) are assumed in T-Matrix calculations (Waterman, 1971; Mishchenko and Travis, 1998). Table 4 shows the comparison of their optical properties: it can be seen that nonspherical particles have a significantly larger lidar ratio $S_p$ whereas the specific extinction cross section $\sigma_e^*$ is nearly unchanged. As a result $\beta_p$ is reduced by 15 - 45 % if nonsphericity is considered, whereas the effect on the AOD is small.

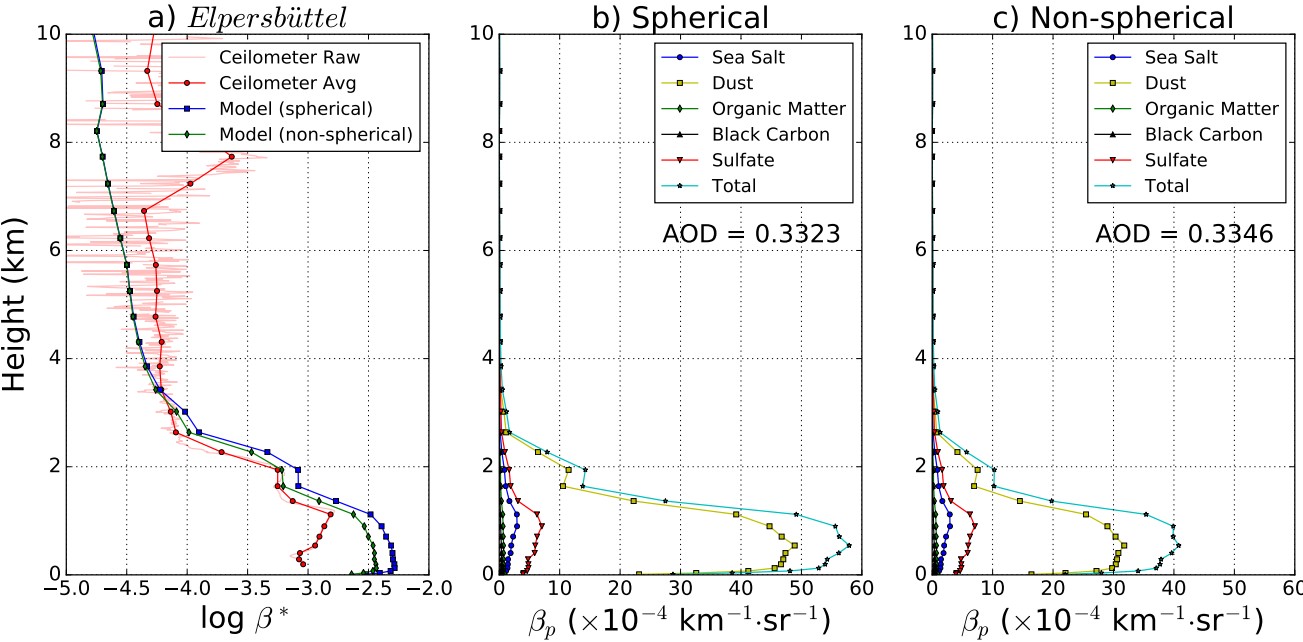

**Figure 8.** (a) Attenuated backscatter derived from the IFS-model and the ceilometer data, respectively, during a dust episode at 18:00 UTC on 3 April 2016 in Elpersbüttel. Model data are converted to $\beta^*$ assuming either spherical (blue curve) and nonspherical (green curve) dust particles. Model results of the particle backscatter coefficient $\beta_p$ (light green) together with the contributions of each aerosol type assuming either spherical (b) or nonspherical (c) particle shape. The aerosol optical depth at 1064 nm is virtually the same (AOD $\approx$ 0.33)

We have also investigated the influence of the treatment of particle shape on the mass to backscatter conversion factors $\eta_\beta$ of the three dust size bins. For demonstration one 1-hour profile from a dust episode (3 April 2016, 18:00 UTC, see also next section) is discussed in detail. Attenuated backscatter profiles are shown in Fig. 8a. Ceilometer measurements with the original vertical resolution of 15 m are shown in light red, whereas the red line shows the ceilometer profile re-sampled for the model's

5      resolution. Profiles derived from the model output for the spherical and nonspherical assumption are given in blue and green, respectively. Fig. 8a clearly demonstrates that the observed decrease of $\beta^*$ in the height range between ∼1.1 km and ∼3.5 km is very well reproduced by the model simulations. However, the absolute values agree somewhat better if nonsphericity is assumed. This improvement is most pronounced in the lowermost layer where dust is the dominating contributor (see Fig. 8b and c): here the overestimate of $\beta^*$ with respect to the ceilometer retrieval is clearly reduced but still in the order of up

10      to a factor of 3 which also implies that the model is overestimating the dust concentration during this episode and/or the aerosol microphysical properties assumed in the forward calculation are different from the actual state. Note, that the increased attenuated backscatter at ∼8 km as observed by the ceilometer is due to the presence of clouds. The modeled $\beta_p$ of the different aerosol types is shown in Fig. 8b and Fig. 8c, assuming either sphericity or nonsphericity of dust particles. Below 3 km dust is by far the dominating aerosol type. As can be expected from Table 4, $\beta_p$ of the dust component is reduced by 15 - 45 % for the

three size bins when nonsphericity is considered. For the profile shown this leads to a reduction of ~33 % of the total particle backscatter coefficient and a better agreement with the observations as shown in the left panel. On the other hand, differences of the aerosol optical depth are negligible (less than 1 %) even during the dust episode. As the concentration of mineral dust aerosol is in general very low in Germany, introducing nonspherical mineral dust in the IFS-model only has a minor impact on the annual average. However, in the case of dust events nonsphericity should be considered to obtain the best possible agreement. This is also expected for volcanic ash layers which are not yet included in the model.

## 4.2 Comparison of the spatiotemporal distribution

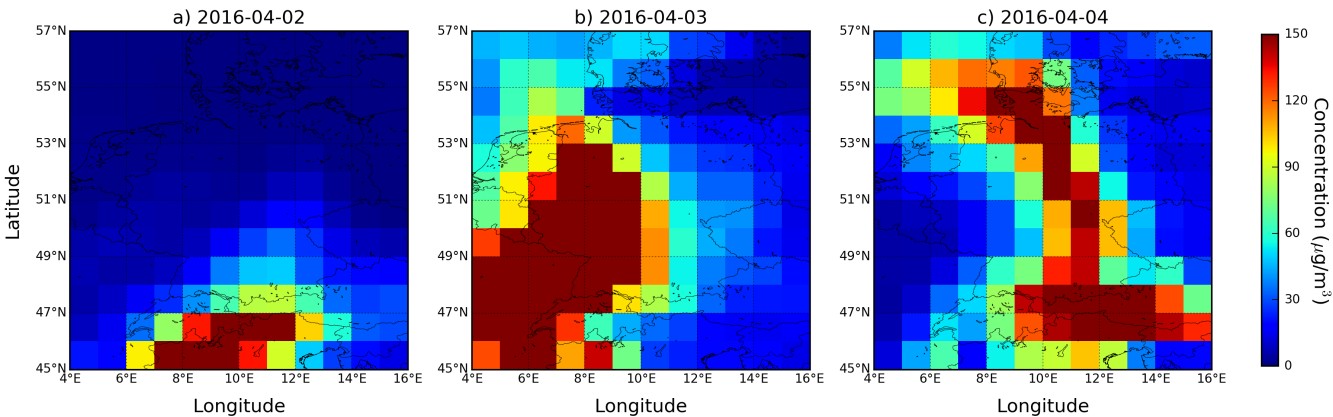

**Figure 9.** Dust concentration (averaged over the lowermost kilometer of the troposphere, in $\mu g/m^3$) over Germany as predicted from the IFS-model: 2 April to 4 April 2016 (from left to right), 12:00 UTC

The focus of the previous section was on the agreement of the attenuated backscatter vertically averaged over the lower troposphere. In the following case study of a dust event we briefly want to outline further options to compare model predictions and measurements of the ceilometer network.

Dust particles are typically a minor contributor to the aerosol abundance in Germany (Beuck et al., 2011; Flentje et al., 2015). On average, it contributes less than 5 % of the total attenuated backscatter according to the IFS model. However, episodes with high concentrations are observed in Germany caused by long range transport of Saharan dust towards Europe (Ansmann et al., 2003; Stuut et al., 2009; Müller et al., 2009; Wiegner et al., 2011). During the one year covered by our study there were two major dust episodes affecting Germany as a whole: in December 2015 and April 2016. The temporal development of the latter from 2 April 2016 to 4 April 2016 is shown in Fig. 9 in terms of the modeled dust concentration (in $\mu g/m^3$), averaged over the lowermost kilometer of the troposphere: the dust layer approached Germany from southwest by 2 April and covered large parts of Germany when moving eastwards (3 and 4 April 2016). The episode came to an end on 5 April when only Austria was still affected. During this event all 12 sites show peak dust contributions of over 50 % of the total $\overline{\beta_{ns}^*}$.

Again we choose Elpersbüttel as an example for the agreement between model and observations. Fig. 10a shows the time-height cross section of the attenuated backscatter of the ceilometer and in Fig. 10b the corresponding profiles calculated from the model output (blue curve) and retrieved from the ceilometer (red curve). Here we treat dust particles as nonspherical particles as defined in Section 4.1.2. Note, that due to cloud filtering some ceilometer profiles stopped at a relatively low
5    altitude.

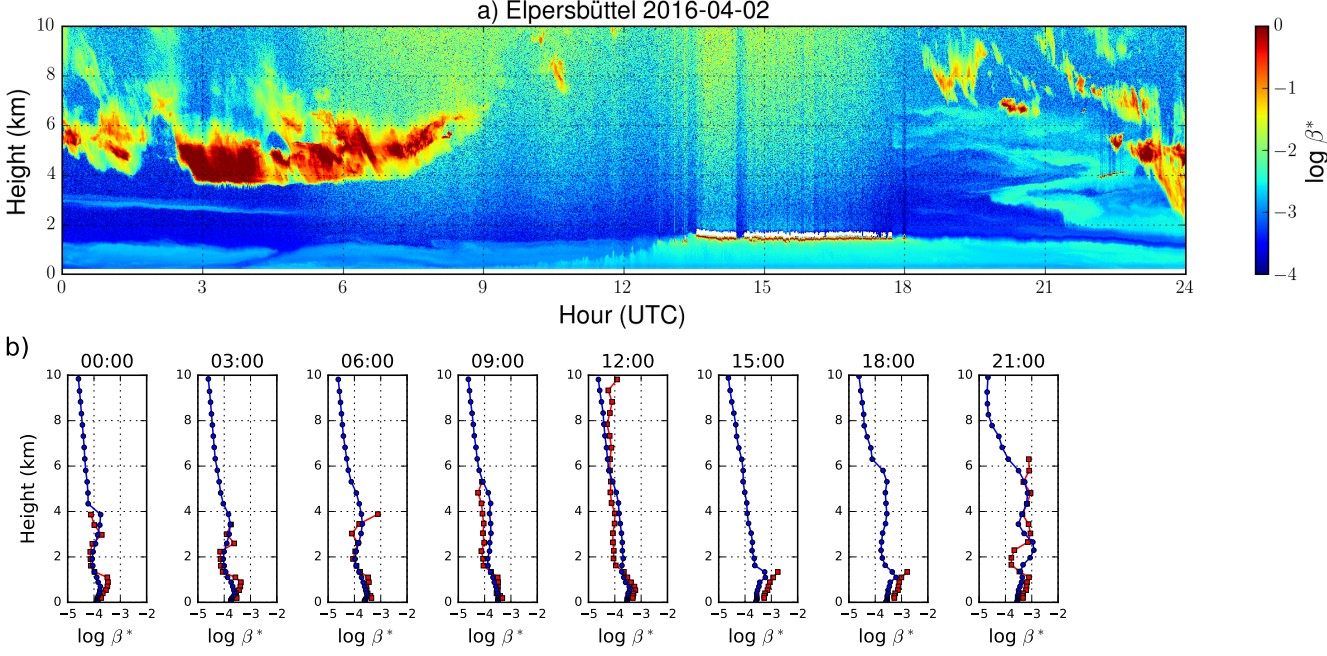

**Figure 10.** Time series of attenuated backscatter measured by the ceilometer at Elpersbüttel during a dust episode on 2 April 2016 is shown in (a). Attenuated backscatter calculated from model simulations (blue curve) is shown in (b), ceilometer measurements (red curve) are averaged to model resolution and shown for reference.

The ceilometer measurements demonstrate that the dust arrived in Elpersbüttel on 2 April 2016 at 18:00 UTC at the latest (light green signatures in Fig. 10a), however, due to the presence of low level clouds the arrival could be up to 4 hours earlier. Pronounced signatures of enhanced backscatter can be observed up to almost 7 km. This is in excellent agreement with the modeled profiles for 18:00 UTC and 21:00 UTC: the aerosol layer is clearly visible up to 6 km and 7 km, respectively; even the
10    pronounced aerosol layer up to approximately 1.5 km is resolved. The absolute values of $\beta^*$ are similar with largest differences in the lowermost kilometer. For the time period before 18:00 UTC the model shows a slightly enhanced $\beta^*$ at altitudes above 3 km, that is not visible in the measurements. On the other hand the vertical extent of the mixing layer is very well reproduced by the model. The enhancement of attenuated backscatter at 4-6 km from 00:00 UTC to 06:00 UTC is due to the presence of clouds.

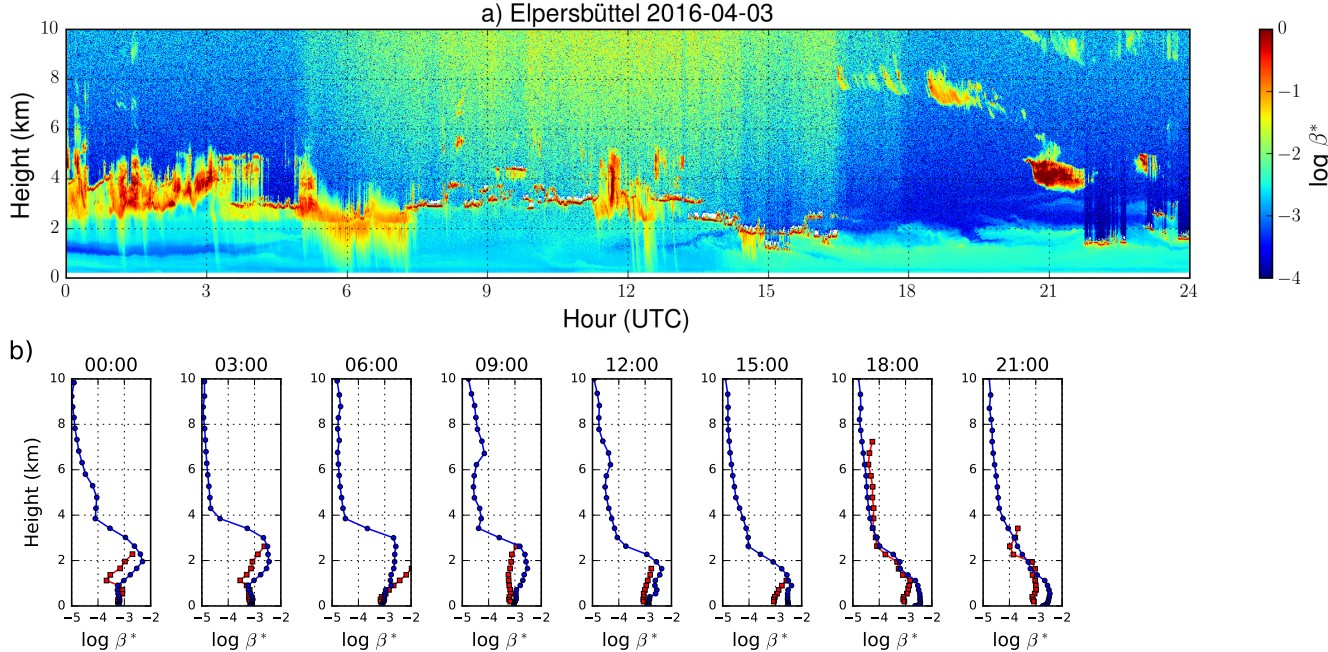

**Figure 11.** Same as Fig. 10 but 3 April 2016.

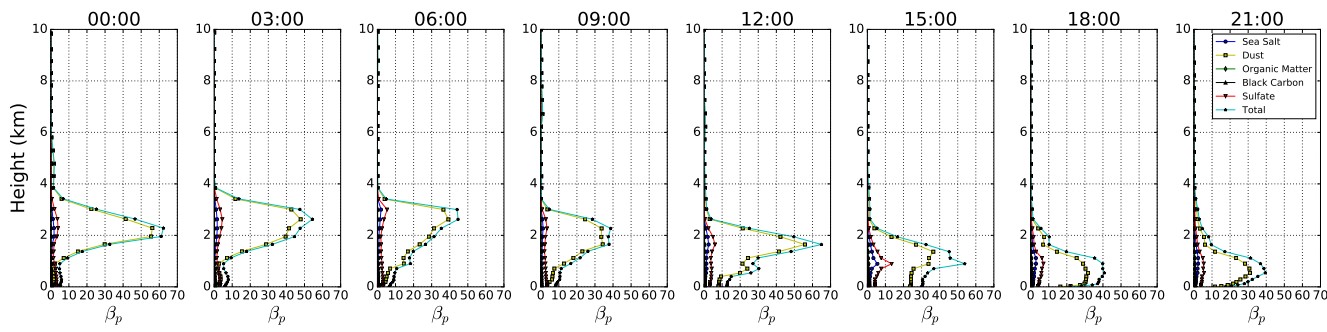

**Figure 12.** Time series of particle backscatter of the five aerosol types simulated by the model at Elpersbüttel during a dust episode on 3 April 2016.

The temporal development of the attenuated backscatter over Elpersbüttel on the following day, 3 April 2016, is shown in Fig. 11, whereas Fig. 12 shows the contribution of particle backscatter for the five aerosol types from the model. The ceilometer measurements show an pronounced elevated aerosol layer which is clearly separated from the surface aerosol layer before 04:00 UTC. From model simulation, the lower layer primarily contains locally produced particles, i.e., sulfate aerosols,

5    whereas the upper layer is (Saharan) dust. This is plausible but cannot be proven from data of a single wavelength backscatter

ceilometer without depolarization channel. Moreover, from the ceilometer data it is not possible to determine the top of the aerosol layer due to clouds, nevertheless measurements at 01:00 UTC and 03:00 UTC suggest that aerosols were present up to approximately 4 km for the first few hours of the night. The model shows large values of $\beta^*$ up to 4 km until 09:00 UTC with dust as the dominating contributor. For the second half of the day the dust layer is confined to the lowermost 3 km according to the model (see Fig. 8). Again, the general agreement of the vertical extent of the aerosol layer is very good. However, it must remain open whether the thin layer at 6 - 7 km, visible in the modeled $\beta^*$-profiles at 09:00 UTC and 12:00 UTC is real or not. The measurement range of the ceilometer is blocked by clouds in 3 km altitude, and even under cloud free conditions the ceilometer might have missed that layer due to the high solar background illumination around noon.

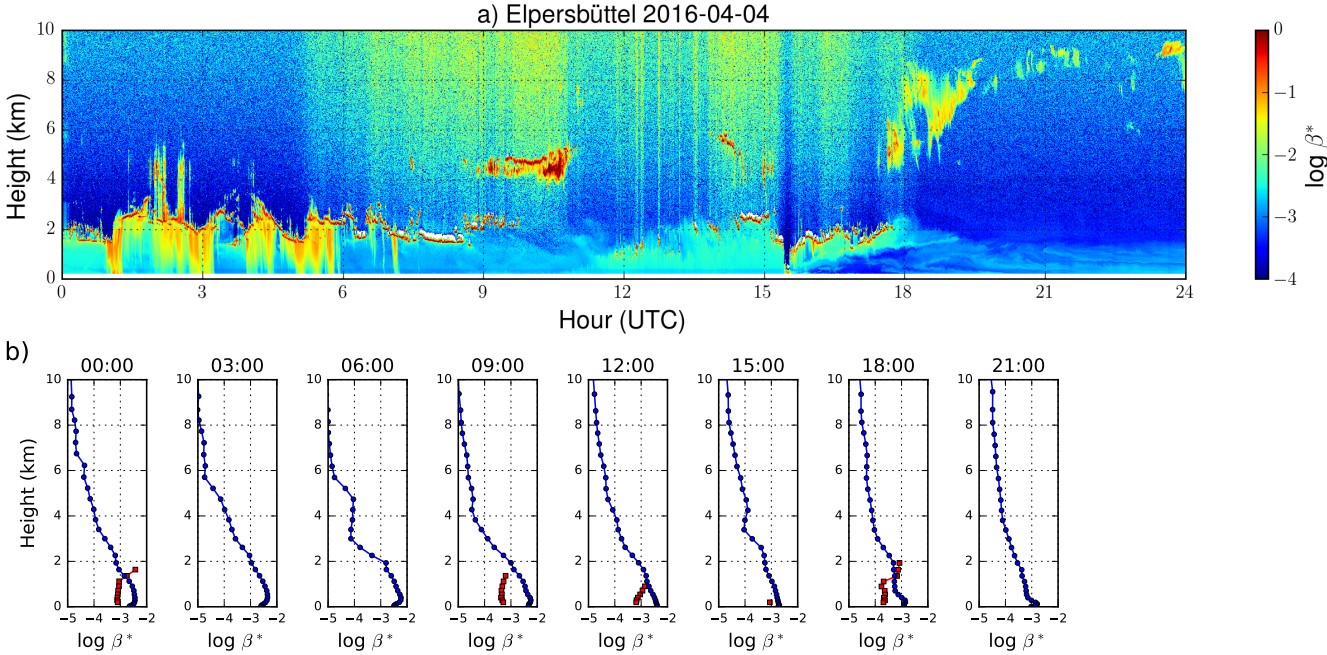

**Figure 13.** same as Fig. 10 but for 4 April 2016.

The situation of the last day of the event is shown in Fig. 13. From the ceilometer observations it can be observed that the elevated aerosol layer disappears at around 19:00 UTC. According to the model simulation the dust event over Elpersbüttel ended on 4 April 2016 at 18:00 - 21:00 UTC. This is in perfect agreement with the ceilometer's attenuated backscatter profile. However, further validation of the vertical extent is hardly possible due to the above mentioned clouds. Again, the upper boundary of the aerosol layer cannot be monitored over the full day, e.g. before 08:00 UTC due to low level clouds. In contrast to the generally good agreement of $\beta^*$ below 1 km, the absolute values differ considerably on 4 April. The discrepancies could be related to the fact, that Elpersbüttel was located at the edge of the high aerosol load region at that time (see Fig. 9c). In this case, misallocation of aerosols in a single grid cell would already result in a huge error. In addition, the local aerosol

distribution certainly had undergone rapid changes due to several rain showers before 07:00 UTC, that might be not resolved by the model.

The overall good agreement between model and observation is confirmed for other sites in Germany. For example, the ceilometer at Soltau, 130 km southeast of Elpersbüttel, observes the dust layer the first time on 2 April, 17:00 UTC, between 3 km and 7 km - in agreement with the model predictions. This also holds for Pelzerhaken (120 km east of Elpersbüttel) where the ceilometer measurements suggests an arrival of the layer by shortly before 22:00 UTC whereas the model results indicate a pronounced dust contribution the first time by 21:00 UTC. Another example is Alfeld, about 250 km south of Elpersbüttel, where the ceilometer observed the arrival of dust layer on 2 April, 17:00 UTC. The dust layer was then gradually descending over night and finally merged into the mixing layer on 3 April, 6:00 - 12:UTC. The time-height cross section of the attenuated backscatter of the ceilometer and the corresponding profiles calculated from the model output over Alfeld from 2. - 4 April are shown in Fig. A1, Fig. A2 and Fig. A3. In central Germany (Offenbach) the arrival time is earlier, approximately at 09:00 UTC according to both model and observations. The upper boundary of the layer is somewhat larger according to the model (6 km vs. 4 km from the observations), however, the ceilometer measurements are subject to high solar background limiting their vertical range. For 3 April the dust event was detected at all stations. In some cases, e.g. Offenbach, the vertical extent of the layer could however not be validated due to low and mid-level clouds.

The case study of the dust episode in April 2016 shows that the model is able to capture such a long range transport event and compare reasonably well with remote sensing measurements. A network of ceilometers is a powerful tool to validate the arrival, the temporal development and the vertical extent of the dust layer as long as low clouds or precipitation do not block the signals. The agreement of the absolute values of $\beta^*$ is however less significant.

## 5   Summary and Conclusions

Numerical simulations of spatio-temporal distribution of aerosols are complex due to manifold interactions between chemistry and meteorology, and the heterogeneity of emission sources. Thus, validation of model forecasts is highly desirable. In this paper, we take advantage of a unique infrastructure: the ceilometer network operated by the German Weather Service (DWD) providing continuous range resolved aerosol information at more than 100 stations. We have compared aerosol model simulation of the European Centre for Medium-Range Weather Forecast Integrated Forecast System (ECMWF-IFS) provided by CAMS with measurements of this ceilometer network. One year of data from September 2015 to August 2016 were considered, and we focus on 12 sites within 20 km of a model grid point. The intercomparison is based on attenuated backscatter $\beta^*$, a quantity that can be derived from well calibrated ceilometers. As the model includes prognostic equations for the mass mixing ratio of 11 different types of aerosols, $\beta^*$-profiles have to be calculated according to the inherent aerosol microphysical properties. Our comparison focus on the lowest part of the atmosphere, i.e. averages $\overline{\beta^*_{ns}}$ from the mean overlap range of the ceilometers at 0.2 km to 1 km above ground. It shows similar annual averages, however, the standard deviation of the difference is in 8 out of 12 sites larger than the average.

To find reasons for the disagreement, we have examined the role of hygroscopic growth of particles and the role of particle shape. We have calculated $\beta^*$ substituting the hygroscopic growth function of sea salt particles based on OPAC by an alternative function reported by Swietlicki et al. (2008). Our calculations show that this change results in a significant reduction of particle backscatter of sea salt. As sea salt is the major contributor to the particle backscatter coefficient, the effect on the modeled attenuated backscatter is in the order of 10 % on average. As a consequence implementing a realistic hygroscopic growth function is essential for the agreement between ceilometer measurements and model.

The importance of an adequate consideration of the nonspherical shape in the case of mineral dust particles was investigated separately. For this purpose calculations of optical properties from the Mie theory and the T-matrix method assuming spheroids were compared. Application of the latter in the framework of a a case study reduces $\beta_p$ of dust by 15 - 45 %, resulting in a better agreement between model and ceilometer measurement. As on average the concentration of dust aerosol is very low in Germany, a significant effect on the total attenuated backscatter is however confined to dust episodes.

Finally we have investigated the 'agreement' between model and observations in the case of a dust event. In this context we understand 'agreement' as the same time period of the event (appearance, dissolution) and the same vertical extent of the dust layer. The case study shows a quite good general qualitative agreement but also highlights the inherent problems of ceilometer measurements when low clouds are present, and the lack of information on the aerosol type due to the single-wavelength concept.

Intercomparisons as described will certainly benefit from a better model resolution and an extension of the ceilometer network. Then, more cases can be found where the distance between a model grid point and a ceilometer site is a few kilometers only. This would strengthen the conclusions. A recent update of the IFS does indeed provide a resolution of $0.5°$, and DWD is continuously extending its ceilometer network. Moreover, attenuated backscatter is included in the model's output since 26 September 2016 facilitating future intercomparisons.

Our study demonstrated that ceilometer networks could offer several options for the validation of numerical models: not only the vertical profile of $\beta^*$, but also the agreement in terms of altitude, extent, temporal development and mean particle backscatter $\beta_p$ of extended/elevated aerosol layers (e.g. volcanic ash) can be considered. In this paper we have discussed only one dust event for demonstration purposes and found good agreement with respect to the vertical extent of the layer and its temporal development. Whether this finding is valid in general must be investigated in further studies. This effort could benefit from the development of automated algorithms for layer detection. Due to their unprecedented spatial coverage ceilometer networks may constitute the observational backbone, nevertheless the combination with supplementary data set, e.g. from advanced lidar systems and photometers for particle characterization, should be fostered.

*Code and data availability.* The source code of the ECMWF IFS model is not available for public as it is an operational model running on routine bases. The ECMWF IFS model simulation results are available to the meteorological offices of the member states of ECMWF. The raw data of the ceilometer instruments are available on request from the data originator DWD (datenservice@dwd.de). The database of aerosol optical properties used in this study is available on request from the corresponding author (ka.chan@dlr.de).

*Competing interests.* The authors declare that they have no conflict of interest.

*Acknowledgements.* We thank Maxime Hervo and his colleagues at MeteoSwiss for the provision of the TOPROF/E-Profile Rayleigh calibration routine. This work has been partially supported by the European Center for Medium Weather Forecast (ECMWF) through its main contractor Royal Netherlands Meteorological Institute (KNMI) and KNMI's subcontractor German Weather Service (DWD) in the context
5  of the Copernicus Atmosphere Monitoring Service (CAMS). The work described in this paper was partially supported by the Marie Curie Initial Training Network of the European Seventh Framework Programme (Grant No. 607905) and the European Cooperation in Science and Technology (COST) action 'TOPROF' of the European Union's Horizon 2020 programme (Project No. ES1303). Josef Gasteiger has received funding from the European Research Council (ERC) under the European Union's Horizon 2020 research and innovation programme (Grant No. 640458, A-LIFE).

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

## Appendix A

The aerosol microphysical and optical properties at ceilometer and lidar applications relevant wavelengths are listed in Table A1. The wavelengths corresponding to Nd:YAG-lasers used for aerosol remote sensing (355 nm, 532 nm and 1064 nm), the widely used Vaisala ceilometers (910 nm), and the 'typical wavelength' for radiative transfer calculations in the shortwave spectral range (550 nm) are shown.

**Table A1.** Microphysical properties of dry aerosols assumed in the model.

| Aerosol Type | Wavelength ($\lambda$, nm) | Density ($\varrho_p$, g/cm$^3$) | Modal Radius (r$_0$, $\mu$m) | Geometric Standard Deviation ($\sigma_g$) | Refractive Index ($n$) | Single Scattering Albedo ($\omega_0$) | Specific Extinction Cross Section ($\sigma_e^*$, m$^2$/g) | Lidar Ratio ($S_p$, sr) |
|---|---|---|---|---|---|---|---|---|
| Sea Salt (bin 1)[a] | 355 | 2.160 | 0.1002,1.0020[c] | 1.9,2.0[c] | 1.5156-0.0002i | 1.00 | 6.37 | 56.9 |
| Sea Salt (bin 2)[a] | 355 | 2.160 | 0.1002,1.0020[c] | 1.9,2.0[c] | 1.5156-0.0002i | 0.99 | 0.57 | 13.5 |
| Sea Salt (bin 3)[a] | 355 | 2.160 | 0.1002,1.0020[c] | 1.9,2.0[c] | 1.5156-0.0002i | 0.97 | 0.16 | 18.8 |
| Dust (bin 1)[b] | 355 | 2.610 | 0.2900 | 2.0 | 1.4800-0.0025i | 0.97 | 2.09 | 16.6 |
| Dust (bin 2)[b] | 355 | 2.610 | 0.2900 | 2.0 | 1.4800-0.0025i | 0.94 | 0.99 | 19.2 |
| Dust (bin 3)[b] | 355 | 2.610 | 0.2900 | 2.0 | 1.4800-0.0025i | 0.89 | 0.40 | 29.5 |
| Organic Matter | 355 | 1.769 | 0.0355 | 2.0 | 1.5280-0.0000i | 1.00 | 5.69 | 35.5 |
| Black Carbon | 355 | 1.000 | 0.0118 | 2.0 | 1.7500-0.4500i | 0.29 | 16.47 | 96.9 |
| Sulfate | 355 | 1.769 | 0.0355 | 2.0 | 1.5280-0.0000i | 1.00 | 5.69 | 35.5 |
| Sea Salt (bin 1)[a] | 532 | 2.160 | 0.1002,1.0020[c] | 1.9,2.0[c] | 1.5156-0.0002i | 1.00 | 3.56 | 76.0 |
| Sea Salt (bin 2)[a] | 532 | 2.160 | 0.1002,1.0020[c] | 1.9,2.0[c] | 1.5156-0.0002i | 0.99 | 0.61 | 14.5 |
| Sea Salt (bin 3)[a] | 532 | 2.160 | 0.1002,1.0020[c] | 1.9,2.0[c] | 1.5156-0.0002i | 0.98 | 0.17 | 15.7 |
| Dust (bin 1)[b] | 532 | 2.610 | 0.2900 | 2.0 | 1.4800-0.0018i | 0.99 | 2.61 | 38.1 |
| Dust (bin 2)[b] | 532 | 2.610 | 0.2900 | 2.0 | 1.4800-0.0018i | 0.96 | 0.88 | 9.5 |
| Dust (bin 3)[b] | 532 | 2.610 | 0.2900 | 2.0 | 1.4800-0.0018i | 0.94 | 0.42 | 23.0 |
| Organic Matter | 532 | 1.769 | 0.0355 | 2.0 | 1.5227-0.0000i | 1.00 | 3.25 | 42.3 |
| Black Carbon | 532 | 1.000 | 0.0118 | 2.0 | 1.7500-0.4500i | 0.21 | 9.84 | 98.7 |
| Sulfate | 532 | 1.769 | 0.0355 | 2.0 | 1.5227-0.0000i | 1.00 | 3.25 | 42.3 |
| Sea Salt (bin 1)[a] | 550 | 2.160 | 0.1002,1.0020[c] | 1.9,2.0[c] | 1.5156-0.0002i | 1.00 | 3.33 | 74.0 |
| Sea Salt (bin 2)[a] | 550 | 2.160 | 0.1002,1.0020[c] | 1.9,2.0[c] | 1.5156-0.0002i | 0.99 | 0.61 | 14.6 |
| Sea Salt (bin 3)[a] | 550 | 2.160 | 0.1002,1.0020[c] | 1.9,2.0[c] | 1.5156-0.0002i | 0.98 | 0.17 | 15.4 |
| Dust (bin 1)[b] | 550 | 2.610 | 0.2900 | 2.0 | 1.4800-0.0016i | 0.99 | 2.63 | 40.9 |
| Dust (bin 2)[b] | 550 | 2.610 | 0.2900 | 2.0 | 1.4800-0.0016i | 0.97 | 0.87 | 9.9 |
| Dust (bin 3)[b] | 550 | 2.610 | 0.2900 | 2.0 | 1.4800-0.0016i | 0.94 | 0.43 | 20.4 |
| Organic Matter | 550 | 1.769 | 0.0355 | 2.0 | 1.5220-0.0000i | 1.00 | 3.07 | 42.5 |
| Black Carbon | 550 | 1.000 | 0.0118 | 2.0 | 1.7500-0.4500i | 0.21 | 9.41 | 99.8 |
| Sulfate | 550 | 1.769 | 0.0355 | 2.0 | 1.5220-0.0000i | 1.00 | 3.07 | 42.5 |
| Sea Salt (bin 1)[a] | 910 | 2.160 | 0.1002,1.0020[c] | 1.9,2.0[c] | 1.5156-0.0002i | 1.00 | 0.89 | 36.0 |
| Sea Salt (bin 2)[a] | 910 | 2.160 | 0.1002,1.0020[c] | 1.9,2.0[c] | 1.5156-0.0002i | 1.00 | 0.63 | 11.6 |
| Sea Salt (bin 3)[a] | 910 | 2.160 | 0.1002,1.0020[c] | 1.9,2.0[c] | 1.5156-0.0002i | 0.99 | 0.17 | 15.9 |
| Dust (bin 1)[b] | 910 | 2.610 | 0.2900 | 2.0 | 1.4800-0.0006i | 1.00 | 1.91 | 74.5 |
| Dust (bin 2)[b] | 910 | 2.610 | 0.2900 | 2.0 | 1.4800-0.0006i | 1.00 | 1.54 | 35.2 |
| Dust (bin 3)[b] | 910 | 2.610 | 0.2900 | 2.0 | 1.4800-0.0006i | 0.98 | 0.41 | 11.8 |
| Organic Matter | 910 | 1.769 | 0.0355 | 2.0 | 1.5114-0.0000i | 1.00 | 1.12 | 37.5 |
| Black Carbon | 910 | 1.000 | 0.0118 | 2.0 | 1.7500-0.4500i | 0.11 | 4.78 | 140.3 |
| Sulfate | 910 | 1.769 | 0.0355 | 2.0 | 1.5114-0.0000i | 1.00 | 1.12 | 37.5 |
| Sea Salt (bin 1)[a] | 1064 | 2.160 | 0.1002,1.0020[c] | 1.9,2.0[c] | 1.5156-0.0002i | 1.00 | 0.55 | 21.7 |
| Sea Salt (bin 2)[a] | 1064 | 2.160 | 0.1002,1.0020[c] | 1.9,2.0[c] | 1.5156-0.0002i | 1.00 | 0.62 | 10.0 |
| Sea Salt (bin 3)[a] | 1064 | 2.160 | 0.1002,1.0020[c] | 1.9,2.0[c] | 1.5156-0.0002i | 0.99 | 0.18 | 18.2 |
| Dust (bin 1)[b] | 1064 | 2.610 | 0.2900 | 2.0 | 1.4800-0.0006i | 1.00 | 1.50 | 78.6 |
| Dust (bin 2)[b] | 1064 | 2.610 | 0.2900 | 2.0 | 1.4800-0.0006i | 1.00 | 1.61 | 48.6 |
| Dust (bin 3)[b] | 1064 | 2.610 | 0.2900 | 2.0 | 1.4800-0.0006i | 0.99 | 0.44 | 13.4 |
| Organic Matter | 1064 | 1.769 | 0.0355 | 2.0 | 1.5068-0.0000i | 1.00 | 0.77 | 34.2 |
| Black Carbon | 1064 | 1.000 | 0.0118 | 2.0 | 1.7500-0.4500i | 0.08 | 3.90 | 168.3 |
| Sulfate | 1064 | 1.769 | 0.0355 | 2.0 | 1.5068-0.0000i | 1.00 | 0.77 | 34.2 |

[a] Sea salt aerosols are represented in the model by three size bins with the bin limits set to 0.015-0.251 $\mu$m (bin 1), 0.251-2.515 $\mu$m (bin 2) and 2.515-10.060 $\mu$m (bin 3).

[b] Dust aerosols are represented in the model by three size bins with the bin limits are set to 0.03-0.55 $\mu$m (bin 1), 0.55-0.90 $\mu$m (bin 2) and 0.90-20.00 $\mu$m (bin 3).

[c] A bimodal lognormal size distribution is assumed for sea salt aerosols, with r$_0$=0.1002 $\mu$m and 1.002 $\mu$m and $\sigma_g$=1.9 and 2.0. The number concentrations $N_1$ and $N_2$ of the first and second mode are 70 and 3 cm$^{-1}$, respectively.

Note that density of hydrophilic aerosol changes with hygroscopic growth of particle.

Fig. A1a shows the time-height cross section of the attenuated backscatter of the ceilometer at Alfeld, ∼250 km south of Elpersbüttel, on 2 April 2016 and in Fig. A1b the corresponding profiles calculated from the model output (blue curve) and retrieved from the ceilometer (red curve). Dust particles are treated as nonspherical particles as defined in Section 4.1.2. Note, that due to cloud filtering some ceilometer profiles stopped at a relatively low altitude. Similar plots for 3 and 4 April 2016 are shown in Fig. A2 and Fig. A3.

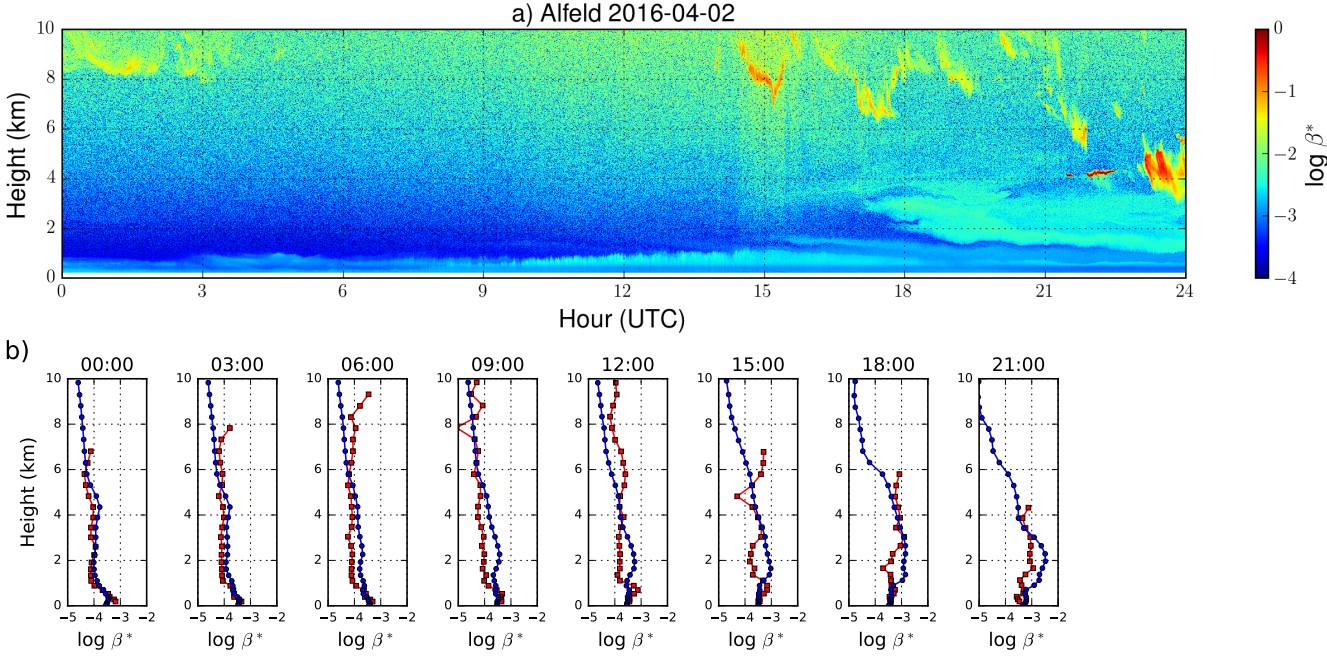

**Figure A1.** Time series of attenuated backscatter measured by the ceilometer (upper panel) and simulated by the model (lower panel) at Alfeld during a dust episode on 2 April 2016.

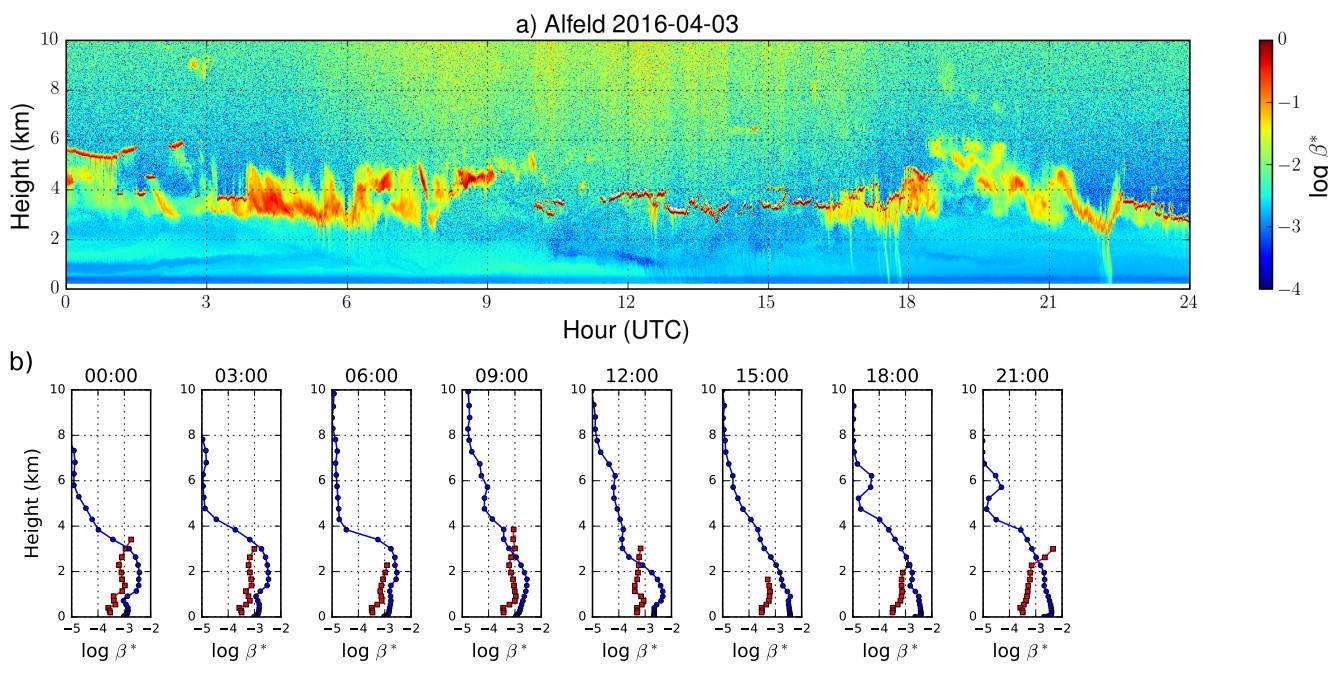

**Figure A2.** Time series of attenuated backscatter measured by the ceilometer (upper panel) and simulated by the model (lower panel) at Alfeld during a dust episode on 3 April 2016.

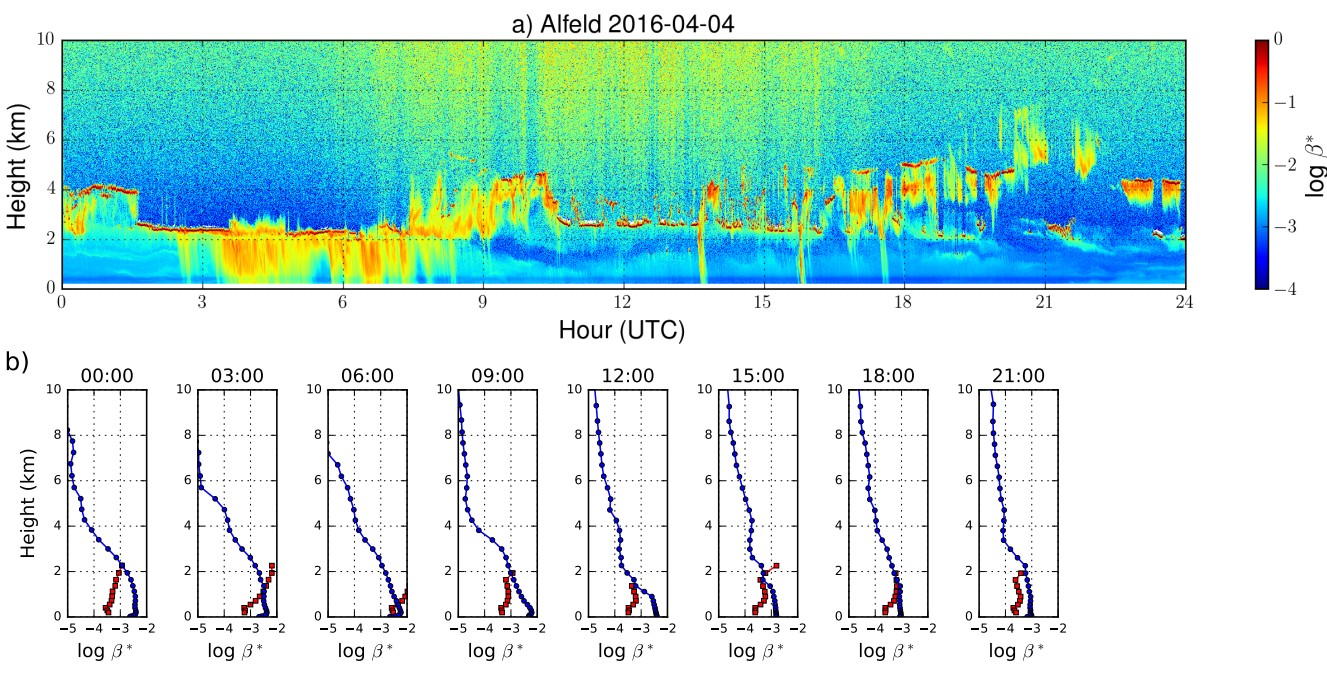

**Figure A3.** Time series of attenuated backscatter measured by the ceilometer (upper panel) and simulated by the model (lower panel) at Alfeld during a dust episode on 4 April 2016.