# Peer review of "Evaluation of ECMWF IFS (version 41R1) operational model forecasts of aerosol transport by using ceilometer network measurements"

_Geoscientific Model Development, 2018_

## Referee Comment (RC1) · Anonymous Referee #1 · 4 May 2018

General comment: The authors have compared attenuated backscatter profiles calculated from model simulation of the European Centre for Medium-Range Weather Forecast Integrated Forecast System (ECMWF-IFS) and ceilometer network measurements operated by the German weather service (DWD) over one year from September 2015 to August 2016. For this comparison it was necessary to convert the mass mixing ratios of 11 aerosols types of the model to attenuated backscatter described in detail in Section 3.1. This conversion involves a lot of assumptions, simplifications and uncertainties, and not surprisingly, the agreement with the ceilometers is not very strong. Given the complexity of the approach and the discrepancies in the results the benefit remains unclear. The ceilometer network in Germany is dense enough (and still in-

creasing) to give a relatively complete picture of the vertical aerosol layering over the country. Although the paper is generally well written I am reluctant to support its publication unless the authors explain more convincingly the purpose of their investigation. Detailed comments: p. 3, line 10: explain GEMS p. 6, lines 8-9: Why does this not apply to ceilometers of DWD? Why discussing $\beta$P when not used? p. 10, line 2 ff: better rename CL (e.g. calibration factor instead of constant) as it is variable p. 11, lines 19-21: re-phrase sentence (grammatically not correct) p. 12, line 15: 120 ceilometer profiles per which time span? p. 15, lines 5-6: is there any proof of this statement? (we learn that the presented IFS model results are very uncertain) Later the authors state that sea salt is probably over-estimated. Fig. 10a: the high backscatter between 00 and 06 UTC is not discussed/explained. p. 15, lines 26-27: The night-time mixing height is very likely even much lower than the mentioned 1.5 km. The phrase in parentheses does not support the statement outside. Section 4.2: partly speculative, many unproven assumptions, not convincing

---

## Short Comment (SC1) · 11 May 2018

Dear authors,

please note, that if only one model is concerned, the title of a GMD manuscript should state the model name (or its acronym) and a version number. These are also important to know in the case of an evaluation, as different versions might perform differently for the same evaluation procedure. Therefore please change the title of your manuscript accordingly upon revision; e.g., Evaluation of ECMWF IFS (v X.Y) operational model forecasts of aerosol transport in using ceilometer network measurements.

[Figure]

Best regards,

Astrid Kerkweg (executive Editor)
* * *

---

## Referee Comment (RC2) · Anonymous Referee #2 · 15 Jun 2018

**General comment**

The paper "Evaluation of operational model forecasts of aerosol transport using ceilometer network measurements" aims at comparing the aerosol distribution forecasted by the CAMS-ECMWF aerosol model to measurements from a ceilometer network operative over Germany. The comparison covers one year (2015-2016). These type of model evaluations are very useful to highlight model errors and to improve the aerosol modelling and given that ceilometers are generally easier and cheaper to maintain than more complex lidar system, they could provide much needed extra informations alongside more complete observations from network such as AERONET.

I think the paper needs mayor revisions before being accepted for publication. The major problem I see are the weak conclusions drawn from the comparison given the relative limited information that can be extracted from the ceilometer signal. Although these shortcomings are somehow acknowledged throughout the paper, a number of speculative conclusions are nevertheless attempted and this makes the overall results of the analysis somewhat unclear. Moreover, the number of assumption needed to compute the attenuated backscatter from the model are not always discussed in detail.

The language is generally clear throughout but it could be improved by some extra polishing.

**Specific comments**

**Abstract** Here and in the rest of the paper: for completeness it should be stated that the aerosol forecasts are produced within the Copernicus Atmosphere Monitoring Service (CAMS) using the aerosol module developed within the GEMS and MACC projects and coupled to the IFS. **line 2**: The comparison is really using the mixing ratio from the IFS, not the backscatter profile, which was not available in the model cycle used in this work. **line 8**: "slightly" too vague, it does not really mean much here. **line 18**: not sure what to make of this: it does not make for a grand introduction to the work and downplays the analysis

**Introduction line 9-14**: about the complexity of atmospheric modelling is perhaps better to provide a short discussion on the current status of aerosol modelling and sources rather than state that it is indeed a difficult problem

**Section 2.1** Here it should be specified that the operational ECMWF forecasts do not provide any aerosol information. Only the forecasts provided by CAMS are produced by coupling an aerosol and chemistry module to the ECMWF IFS to provide analysis and forecasts of atmospheric composition It is not clear from this section which data are used in the comparison. Is it analysis fields? Or forecasts? If forecasts, at which lead time? **pag4, line 22**: given that only results for wavelengths relevant for ceilometers
are discussed, there is not point to show values for other wavelengths here. Also, the table could be restructured using two columns per optical property to show the values for the relevant wavelength, eliminating the need of copy and paste all the other informations for each wavelengths. pag4, line 27: the horizontal resolution should be a Gaussian grid, not regular. The CAMS forecasts for CY41R1 should be at a spectral truncation TL255, roughly equivalent to a 0.7x0.7 degrees resolution. Please check the information. pag4, line 29: in cy41R1 aerosol in IFS are not interactive with radiation and no explicit output of backscatter profile is provided. Hence the information about the assumptions in the optical properties used in IFS are not relevant here. Given that the computation of the backscatter profile is done off line using the aerosol mixing ratio from the model, the choices of refractive index and size distribution is entirely up to the user. The choices should be discussed in a separate sub section, and if the user wants to adopt the same values used in the IFS for the computation of the aerosol optical depth, it should be justified. Also this is the place to discuss further choices in the treatment of optical properties (e.g. hydrophilic growing factors and particle shape) pag4, line 26: modal radius and limits of integration over the size distribution pag 5, line 4: this has to be explained a bit more carefully because it might be relevant given the results shown later on.

section 3.1 Not clear: the title of the section says attenuated backscatter but from the text it looks like the computed quantity here is the true layer backscatter. pag 7, line 20: unusual terminology, isn't it equation 7 just the definition of the mass extinction coefficient?

section 3.2 pag9, line 2: define slow. Will impact a full year of data like in this work?
pag 10, line 10: 'sky-condition-index' and 'cloud-base-height' not defined. Not clear how they are used, is it to exclude data not relevant for aerosol comparisons? pag 10, line 11: If mentioned it is probably useful to have an idea of how much this variation in the accuracy of the calibration constant actually is.

section 4 pag 10, lines 19-25: not clear pag 10, line 27: as already outlined: not
clear which model data have been used. Forecast fields? Analysis? **pag 10, line 29**: confusing, why here 2 km maximum height is used and few lines before 1 km was mentioned?

**section 4.1 pag 11, line 10**: well perhaps a look at some of those situations might help to give some clue. Aren't the events in December and at the beginning of April 2016 the dust advection cases discussed later on? **pag 11, line 15**: how does it compare to the uncertainty expected from the measurements at each site? Perhaps a table with the annual mean and some measure of uncertainty and dispersion of the data at each site gives a clearer picture. **pag12, line 1**: here it is meant larger or smaller than sigma in absolute value I guess **pag 12, line 21-25**: if it is the case that sea salt is largely overestimated, there should be a discussion showing the contribution of all aerosol types to the total AOD and total mass for each site, not only the contribution to the backscatter.

**section 4.1.1** this is really relevant only if the influence of the overestimation in total sea salt amount and in the choices of optical properties are not the main reason behind the discrepancy (which most likely are it seems). Moreover given the difficulties highlighted throughout the test (e.g. pag 13 line 10) and the relatively small contribution that this correction brings, this section could be significantly reduced.

section 4.1.2 pag 13, line 26: not necessarily. Non-sphericity might have a nonnegligible contribution to the lidar backscatter signal, but for flux computations, e.g. in a typical radiation code of a climate or NWP model, the impact is often small (e.g. Räisänen et al. 2012 https://doi.org/10.1002/qj.2084) pag 14, line 7: I think that it's clear that the vertical profile is not affected by the choice of particle shape. pag 14, line 11: The choice of size distribution/refractive index also plays a role.

section 4.2 pag 15, line 14: it could be nice to see another one or two sites since Elpersbuettel is at the edge of the event and more susceptible to errors in the plume location. pag 15, line 16: why? from the IFS only the mass mixing ratio is used, there is
no need to be consistent with other assumptions here. If the non spherical assumption brings results slightly closer to the observations, then perhaps this should be used. pag 15, line 17-18: "it seems". It should be discussed better pag 15, line 19: plotting the two profiles (model-observed) on the same chart will help the comparison pag 15, line 26: why this assumption if it cannot be proven? pag 15, line 31: again, perhaps showing the model profile broken down in the 5 aerosol species cloud help pag 15, lines 33-34 pag 16, lines 1-2: too speculative, does not add to the general discussion. pag 16, line3: not easy to see from the plot. pag 16, line 4-6: from the ceilometer alone not much can really be said. Does the model speciation show the decrease in dust mixing ratio? **pag 16, lines 8-11**: guite speculative and not much relevant **pag** 16, lines 11-20: it could be interesting to see it. Otherwise there is not much point in mentioning it. pag 16, lines 21-26: rather inconclusive paragraph. If the discussion would stick to what can be seen from the ceilometer without trying to extrapolate too much beyond (probable hieght above cloud layers, uncertain arrival and dissolution of the aerosol plume, speciation). I think the interesting result to highlight is that the main feature of such an event can be captured and compare reasonably well with the model fields.

---

## Author Comment (AC1) · 13 Aug 2018

We thank reviewer #1 for the quick response and the comments. These comments were helpful for improving our manuscript. We have addressed the reviewer's comments on a point to point basis as below for consideration.

**General comment:**

The authors have compared attenuated backscatter profiles calculated from model simulation of the European Centre for Medium-Range Weather Forecast Integrated Forecast System (ECMWF-IFS) and ceilometer network measurements operated by the

German weather service (DWD) over one year from September 2015 to August 2016. For this comparison it was necessary to convert the mass mixing ratios of 11 aerosols types of the model to attenuated backscatter described in detail in Section 3.1. This conversion involves a lot of assumptions, simplifications and uncertainties, and not surprisingly, the agreement with the ceilometers is not very strong. Given the complexity of the approach and the discrepancies in the results the benefit remains unclear. The ceilometer network in Germany is dense enough (and still increasing) to give a relatively complete picture of the vertical aerosol layering over the country. Although the paper is generally well written I am reluctant to support its publication unless the authors explain more convincingly the purpose of their investigation.

Response: The main concern of the reviewer is the large number "of assumptions, simplifications and uncertainties" used in the conversion procedure. We don't feel that this criticism is justified, otherwise, remote sensing (from ground and space) cannot be used for validation purposes at all. As remote sensing is relying on measurements of components of the radiation field, and models provide primarily physical properties (e.g. mass mixing ratios) conversion is an intrinsic feature of this kind of validation or comparison. In general the conversion of physical properties to optical properties is much better defined than the opposite direction, thus, in our case the uncertainties are comparably small: scattering theory (Mie or T-matrix approach) is generally accepted and if the microphysical properties are known (in our case prescribed or calculated from the model) the conversion is "exact". Open questions are discussed in our manuscript: Is the assumption of spherical particles correct and what is its influence on the "reality" (in our case the measurements)? Is the choice of the hygroscopic growth model relevant? In this context we have described the theoretical background and the inherent assumptions, so that the reader can understand what was done.

To determine the agreement or disagreement between observations and model output was the goal of our study. A key point was to find out if improvements with respect to the modeling of the hygroscopic growth and the consideration of particle shape can reduce

the disagreement. We believe that this is a clear benefit of the study as it could help to create sort of a priority list for modifications of the model physics (is it worthwhile to spend efforts on a certain topic?). Moreover, as to our knowledge the use of the ceilometer network (an already existing routinely 24/7 working infrastructure) for these purposes has not been investigated before our paper can be a first step towards new applications of this infrastructure.

Detailed comments:

p. 3, line 10: explain GEMS

Response: We have now supplemented the information of GEMS (page 1, line 4; page 4, line 10-11).

p. 6, lines 8-9: Why does this not apply to ceilometers of DWD? Why discussing  $\beta p$  when not used?

Response: It is because ceilometers of the DWD measure at 1064 nm which is not affected by the water vapor absorption. Nevertheless this topic should be mentioned as the majority of ceilometers are operating in the water vapor absorption band. A detailed description of the DWD ceilometers network is presented in section 2.2. We have recapped the measurement wavelength of DWD ceilometers to avoid confusion. In addition, we have removed the information of  $\beta p$  error to avoid redundancy (page 9, line 13-14).

p. 10, line 2 ff: better rename CL (e.g. calibration factor instead of constant) as it is variable

Response: In the lidar community the term "lidar constant" is common, whereas operators of ceilometers use both terms synonymously. We agree that "calibration factor" better indicates that the value might be time-dependent. We believe that all scientists working in the lidar-field are aware of this fact even if they use "lidar constant". Nevertheless we have clarified the reviewer's concern in the revised manuscript (page 9, line

7-8).

p. 11, lines 19-21: re-phrase sentence (grammatically not correct)

Response: We have rephrased the sentence (page 14, line 9-10).

p. 12, line 15: 120 ceilometer profiles per which time span?

Response: Individual ceilometer profile is taken every 15s. In this study, we compare hourly averaged ceilometers data to model simulation. As a consequence, averages consider 240 ceilometer profiles at maximum. On the other hand, data contaminated by low clouds and precipitation are not considered in this study. The total least squares regression line is based only on intercomparisons when the hourly averaged data contains at least 120 ceilometer profiles (30 minutes of measurements). We have rephrased the sentence to avoid confusion (page 16, line 2-3).

p. 15, lines 5-6: is there any proof of this statement? (we learn that the presented IFS model results are very uncertain) Later the authors state that sea salt is probably over-estimated. Fig. 10a: the high backscatter between 00 and 06 UTC is not discussed/explained.

Response: We have now included references to support the fact that dust particle is a minor contributor to the aerosol abundance in Germany (page 22, line 11). The high backscatter from 00:00UTC to 06:00UTC is due to the present of cloud. We have supplemented the explanation on page 23, line 13-14.

p. 15, lines 26-27: The night-time mixing height is very likely even much lower than the mentioned 1.5 km. The phrase in parentheses does not support the statement outside.

Response: We mean that the maximum height (i.e. in the afternoon) of the mixing layer is usual below 1.5km during spring time. Of course, the mixing layer height could be lower during night time, but it does not contradict the statement. We have rephrased the sentence to avoid confusion (page 24, line 4-5).

Section 4.2: partly speculative, many unproven assumptions, not convincing.

Response: We have revised the section and provided more details from both model simulation and ceilometer measurement. In addition, we have rephrased the section to make it less speculative.

---

## Author Comment (AC2) · 13 Aug 2018

We thank reviewer #2 for careful reading our manuscript and the very detailed and helpful comments. They certainly helped us to improve the manuscript. We understand that the comments on the scientific content of the manuscript in general are positive, however, several clarifications are necessary. We hope the revised form of the manuscript has improved in all aspects and the manuscript is relevant to aims and scope of the journal. We have addressed the reviewer's comments on a point to point basis as below for consideration.

General comment

The paper "Evaluation of operational model forecasts of aerosol transport using ceilometer network measurements" aims at comparing the aerosol distribution forecasted by the CAMS-ECMWF aerosol model to measurements from a ceilometer network operative over Germany. The comparison covers one year (2015-2016). These type of model evaluations are very useful to highlight model errors and to improve the aerosol modeling and given that ceilometers are generally easier and cheaper to maintain than more complex lidar system, they could provide much needed extra information alongside more complete observations from network such as AERONET.

I think the paper needs mayor revisions before being accepted for publication. The major problem I see are the weak conclusions drawn from the comparison given the relative limited information that can be extracted from the ceilometer signal. Although these shortcomings are somehow acknowledged throughout the paper, a number of speculative conclusions are nevertheless attempted and this makes the overall results of the analysis somewhat unclear. Moreover, the number of assumption needed to compute the attenuated backscatter from the model are not always discussed in detail.

The language is generally clear throughout but it could be improved by some extra polishing.

**Specific comments**

Abstract Here and in the rest of the paper: for completeness it should be stated that the aerosol forecasts are produced within the Copernicus Atmosphere Monitoring Service (CAMS) using the aerosol module developed within the GEMS and MACC projects and coupled to the IFS.

Response: We have now included a better description of the model output used in this study (abstract and section 2.1).

line 2: The comparison is really using the mixing ratio from the IFS, not the backscatter profile, which was not available in the model cycle used in this work.

Response: We have now changed "aerosol backscatter profile" to a generic "aerosol profile" (a precise explanation is provided later in the paper) in order to avoid confusion. A more detailed description of the forward calculation for the conversion of aerosol mixing ratios to backscatter profiles is presented later in the manuscript.

line 8: "slightly" too vague, it does not really mean much here.

Response: We removed the word 'slightly' from the sentence (page 1, line 10).

line 18: not sure what to make of this: it does not make for a grand introduction to the work and downplays the analysis

Response: We removed the sentence from the abstract (page 2, line 1).

Introduction line 9-14: about the complexity of atmospheric modelling is perhaps better to provide a short discussion on the current status of aerosol modelling and sources rather than state that it is indeed a difficult problem

Response: We have revised the introduction to include a selection of relevant citations concerning the current status of aerosol modeling aerosol emission sources, and comparisons with observations (page 2, line 15 to page 3, line 11).

Section 2.1 Here it should be specified that the operational ECMWF forecasts do not provide any aerosol information. Only the forecasts provided by CAMS are produced by coupling an aerosol and chemistry module to the ECMWF IFS to provide analysis and forecasts of atmospheric composition It is not clear from this section which data are used in the comparison. Is it analysis fields? Or forecasts? If forecasts, at which lead time?

Response: In order to avoid the confusion, we have now revised the description and state clear that the aerosol simulation is provided by CAMS with the coupling of the aerosol and chemistry module to the ECMWF-IFS model. Daily forecast data are taken at 00:00 UTC each day, resulting a forecast lead time of 0-21 hours. This information is now included in the manuscript (page 4, line 9-10).

СЗ

pag4, line 22: given that only results for wavelengths relevant for ceilometers are discussed, there is not point to show values for other wavelengths here. Also, the table could be restructured using two columns per optical property to show the values for the relevant wavelength, eliminating the need of copy and paste all the other information for each wavelengths.

Response: Although some of the wavelengths shown in the manuscript are not used in this study, they are relevant for other common ceilometers and aerosol lidar applications. Thus we believe that this information might be useful. Nevertheless, we followed the reviewer's comment and removed the other wavelengths from the description. In addition, we have also modified Table 2 and put the information of the other wavelengths in appendix (Table A1).

pag4, line 27: the horizontal resolution should be a Gaussian grid, not regular. The CAMS forecasts for CY41R1 should be at a spectral truncation TL255, roughly equivalent to a 0.7x0.7 degrees resolution. Please check the information.

Response: IFS is a spectral model, but in the Meteorological Archival and Retrieval System (MARS) at ECMWF, GRIB data is archived in one of the following spatial coordinate systems: Spherical Harmonics (SH), Gaussian Grid (GG) or Latitude/Longitude (LL). From this archive we retrieved the data as NetCDF files on a regular lat/lon grid. While the original model resolution is approximately 0.7° (360°/2/255), within MARS, the data is then transformed to a regular grid of 1°x1°. We have supplied the additional information in the manuscript (page 4, line 27-30).

pag4, line 29: in cy41R1 aerosol in IFS are not interactive with radiation and no explicit output of backscatter profile is provided. Hence the information about the assumptions in the optical properties used in IFS are not relevant here. Given that the computation of the backscatter profile is done off line using the aerosol mixing ratio from the model, the choices of refractive index and size distribution is entirely up to the user. The choices should be discussed in a separate sub section, and if the user wants to adopt the same values used in the IFS for the computation of the aerosol optical depth, it should be justified. Also this is the place to discuss further choices in the treatment of optical properties (e.g. hydrophilic growing factors and particle shape)

Response: The optical properties of aerosol are calculated offline from the model output of aerosol mass mixing ratios by assuming aerosol microphysical properties. Some of these properties are defined in the model, i.e. size bins, other properties are taken from external databases. We decided to adapt the aerosol microphysical properties used for aerosol optical depth calculation in the previous study (Morcrette et al., 2009), as it provide a rather complete overview and the resulting aerosol optical depths also agree well with measurements. In addition, we also performed some sensitivity tests, e.g. concerning the hydroscopic growth and the effect of nonspherical particles, to investigate the effects on especially lidar related aerosol optical properties. We follow the reviewer's comment and moved the description of aerosol microphysical properties in new section (section 2.2).

pag4, line 26: modal radius and limits of integration over the size distribution

Response: Revised according to reviewer's comment (page 5, line 32).

pag 5, line 4: this has to be explained a bit more carefully because it might be relevant given the results shown later on.

Response: A more detailed explanation is included in the manuscript (page 6, line 10-11).

section 3.1 Not clear: the title of the section says attenuated backscatter but from the text it looks like the computed quantity here is the true layer backscatter.

Response: Obviously this section was confusing so that we emphasize at the end of Section 3.1 (page 11, line 8-10) that attenuated backscatter is calculated. Input for this calculation is - among others - the particle backscatter coefficient. The definition of attenuated backscatter is provided in Eq. 2.

pag 7, line 20: unusual terminology, isn't it equation 7 just the definition of the mass extinction coefficient?

Response: In principle the reviewer is right: This is indeed a "mass extinction coefficient". On the other hand a "mass extinction coefficient" usually refers to the "total" size distribution of the particles, whereas here a size range according to the specific bins is considered. So, different mass extinction coefficients are existing. As readers also might be used to that term we have followed the reviewer's suggestion and revised the terminology (page 10, line 16-17).

section 3.2 pag9, line 2: define slow. Will impact a full year of data like in this work?

Response: The DWD ceilometers are calibrated routinely whenever possible (i.e., adequate weather conditions are prevailing). Thus, changes are monitored and can be considered. Details of the calibration are given in the following paragraph (page 12, line 1-12). In this context it is indeed irrelevant whether these changes are "slow" or not. Consequently we rephrased this sentence. (page 11, line 21-22).

pag 10, line 10: 'sky-condition-index' and 'cloud-base-height' not defined. Not clear how they are used, is it to exclude data not relevant for aerosol comparisons?

Response: Those are data quality flags provided by the proprietary software of the ceilometers which are used to filter data contaminated by rain, fog, snow and low level clouds. The definitions of those flags are now provided in the manuscript (page 12, line 16 to page 13, line 2).

pag 10, line 11: If mentioned it is probably useful to have an idea of how much this variation in the accuracy of the calibration constant actually is.

Response: The lidar constant CL is routinely calibrated as mentioned in the previous paragraph. Therefore, possible changes with time can be observed. The typical error of individual calibration is 15-20 %, while the actual error is smaller due to the temporal smoothing. The monthly variation of CL is usually less than 5 % and the annual

variation is 10-15 %. This information is now included in the manuscript (page 12, line 10-12).

section 4 pag 10, lines 19-25: not clear

Response: We have rephrased the corresponding paragraph: our message is that even if  $\beta p$  of an elevated layer agrees, attenuated backscatter may disagree if there are differences (model vs. observations) in the atmosphere below(page 13, line 8-14).

pag 10, line 27: as already outlined: not clear which model data have been used. Forecast fields? Analysis?

Response: This issue has been addressed according to earlier comment.

pag 10, line 29: confusing, why here 2 km maximum height is used and few lines before 1 km was mentioned?

Response: We compare  $\beta \hat{a} \hat{L} \hat{U}$  averaged from 0.2 km to 1 km, while the cloud filtering criterion of low level cloud is '2 km'. In principle the reviewer is right: it would be sufficient to exclude measurements with clouds in the range where we determined  $\beta^*$ . However, to be on the safe side (in the case of errors of the cloud bottom height) we used a cloud filter criterion of 2 km instead of 1 km. Moreover, this criterion allows us to use the same data sets for intercomparisons of profiles as discussed later in the paper. These profiles should at least have a vertical extent of 2 km, otherwise their benefit for aerosol studies is in general limited (e.g., radiation budget). As a consequence, we have rephrased the sentence to avoid confusion (page 13, line 19).

section 4.1 pag 11, line 10: well perhaps a look at some of those situations might help to give some clue. Aren't the events in December and at the beginning of April 2016 the dust advection cases discussed later on?

Response: We have now referred the readers to section 4.1.2 and 4.2 for more detailed analysis (page 13, line 31).

pag 11, line 15: how does it compare to the uncertainty expected from the measurements at each site? Perhaps a table with the annual mean and some measure of uncertainty and dispersion of the data at each site gives a clearer picture.

Response: We followed the reviewer's comment and added a table to summarize the annual mean and the uncertainty of all sites (Table 3).

pag12, line 1: here it is meant larger or smaller than sigma in absolute value I guess

Response: It means the absolute difference between model and observation smaller or larger than the standard deviation obtained from the statistic. The standard deviation of the difference for each site is included in the new Table 3. In the revised version the notation for absolute values has been added (page 14, line 15).

pag 12, line 21-25: if it is the case that sea salt is largely overestimated, there should be a discussion showing the contribution of all aerosol types to the total AOD and total mass for each site, not only the contribution to the backscatter.

Response: Here we just listed some possible reasons for the discrepancy between model and observation. It can be related to the assumed optical properties of aerosols or uncertainty related to the emission and transportation of aerosols in the model. As the focus of the manuscript is the comparison of backscatter data, we have only added a brief summary of the contribution of sea salt with respect to the AOD (following the reviewer's suggestion). The annual averaged sea salt contribution to the total AOD is ranging from 21% (Görlitz) to 37% (Elpersbüttel). The information is supplemented to the manuscript (page 17, line 4 to page 18, line 3).

section 4.1.1 this is really relevant only if the influence of the overestimation in total sea salt amount and in the choices of optical properties are not the main reason behind the discrepancy (which most likely are it seems). Moreover given the difficulties highlighted throughout the test (e.g. pag 13 line 10) and the relatively small contribution that this correction brings, this section could be significantly reduced.

Response: We understand the reviewer's concern that the model errors might have a larger impact on the comparison. However, we could not solve this from the forward operator perspective. Therefore, we tried to quantify another major source of error - hydroscopic growth. As we have shown in the manuscript the influence of using a better hydroscopic growth database would result in a 22% reduction of sea salt backscatter. Considering sea salt contributes over 50% of the total backscatter, a 22% reduction of sea salt backscatter would reduce the total backscatter by more than 10%. With these information the readers can judge by themselves which priority they give to this topic when thinking about improvements of the model, so we think this section is useful. Nevertheless, in response to the reviewer's comment, we have reduced the discussion in this section.

section 4.1.2 pag 13, line 26: not necessarily. Nonsphericity might have a non negligible contribution to the lidar backscatter signal, but for flux computations, e.g. in a typical radiation code of a climate or NWP model, the impact is often small (e.g. Räisänen et al. 2012 https://doi.org/10.1002/qj.2084)

Response: The statement of the reviewer is in agreement to our manuscript (page 20, line 19-20). We have explicitly mentioned that the nonsphericity of particles has only a small impact on the extinction. However, the effect on backscatter is quite large (up to 45%). Insofar the first sentence should not be misunderstood as "important role for all optical properties". We changed this to "lidar related optical properties" (page 20, line 12-13).

pag 14, line 7: I think that it's clear that the vertical profile is not affected by the choice of particle shape.

Response: We removed the sentence 'independent of the numerical treatment of the particle shape' (page 21 line 7).

pag 14, line 11: The choice of size distribution/refractive index also plays a role.

Response: We have now supplemented the information of other possible influences (page 21 line 10-11).

section 4.2 pag 15, line 14: it could be nice to see another one or two sites since Elpersbuettel is at the edge of the event and more susceptible to errors in the plume location.

Response: We have added another example of Alfeld, 250 km south of Elpersbüttel, to illustrate the arrival and the temporal development of the dust episode influences (page 26 line 7-11). Due to the length of the manuscript, we have moved these plots to the appendix.

pag 15, line 16: why? from the IFS only the mass mixing ratio is used, there is no need to be consistent with other assumptions here. If the non spherical assumption brings results slightly closer to the observations, then perhaps this should be used.

Response: The nonspherical assumption does not show a significant impact on the relative attenuated backscatter profile. Therefore, it does not affect the interpretation of the dust layer. However, we follow the reviewer's comment and now used the non-spherical assumption for these plots (page 23, line 3-4 and Fig. 10-13).

pag 15, line 17-18: "it seems". It should be discussed better

Response: We have rephrased the wording to make it less speculative (page 23, line 6).

pag 15, line 19: plotting the two profiles (model-observed) on the same chart will help the comparison

Response: We followed the reviewer's comment and include the averaged ceilometer attenuated backscatter profiles in Fig. 10b, 11b and 13b.

pag 15, line 26: why this assumption if it cannot be proven?

Response: This is not an assumption but a conclusion from the model result. We have

rephrased the sentence to avoid confusion (page 34, line 4-5).

pag 15, line 31: again, perhaps showing the model profile broken down in the 5 aerosol species cloud help

Response: The time series of the contribution of particle backscatter for the five aerosol types is shown in a new Fig. 12 for a better interpretation of temporal development of the dust layer.

pag 15, lines 33-34 pag 16, lines 1-2: too speculative, does not add to the general discussion.

Response: In this point we disagree with the reviewer: We believe that it is important to show the limits of the validation. In the case of low level clouds that cannot be penetrated by the ceilometer measurements, information of the atmosphere above the clouds is not available. This is an inherent problem of all lidar/ceilometer measurements.

pag 16, line3: not easy to see from the plot.

Response: We have now revised the figures and show both ceilometer and model profile in the same plot (Fig. 10b, 11b and 13b.).

pag 16, line 4-6: from the ceilometers alone not much can really be said. Does the model speciation show the decrease in dust mixing ratio?

Response: The model simulation shows the dust concentration gradually decreased during the day and finally disappeared at 18-21 UTC. As there are already too many figures in the manuscript, we decide not showing the aerosol speciation particle backscatter profile. However, we have revised the sentence to avoid any confusion (page 25, line 9-12).

pag 16, lines 8-11: quite speculative and not much relevant

Response: We have removed these sentences from the manuscript.

pag 16, lines 11-20: it could be interesting to see it. Otherwise there is not much point in mentioning it.

Response: We have now included the measurement and model simulation results from Alfeld, about 250km south of Elpersbüttel, to illustrate the arrival and the temporal development of the dust episode. Due to the length of the manuscript, we have moved these plots to the appendix.

pag 16, lines 21-26: rather inconclusive paragraph. If the discussion would stick to what can be seen from the ceilometer without trying to extrapolate too much beyond (probable hieght above cloud layers, uncertain arrival and dissolution of the aerosol plume, speciation), I think the interesting result to highlight is that the main feature of such an event can be captured and compare reasonably well with the model fields.

Response: We have revised the whole paragraph to avoid over interpret the model and observation data (page 26, line 16-19).

---

## Author Comment (AC3) · 13 Aug 2018

please note, that if only one model is concerned, the title of a GMD manuscript should state the model name (or its acronym) and a version number. These are also important to know in the case of an evaluation, as different versions might perform differently for the same evaluation procedure. Therefore please change the title of your manuscript accordingly upon revision; e.g., Evaluation of ECMWF IFS (v X.Y) operational model forecasts of aerosol transport in using ceilometer network measurements.

Response: We thank the editor for the reminder. We have revised the title of the manuscript to 'Evaluation of ECMWF IFS (version 41R1) operational model forecasts

of aerosol transport by using ceilometer network measurements' in order to fulfill the requirement of the journal.

---

## Author Response (AR2)

**Response to reviewer**

We thank reviewer for careful reading our manuscript. The comments raised by the reviewer are certainly helpful to improve the manuscript. The reviewer in general appreciates our effort in revising the manuscript, however, several technical corrections are necessary. We have carefully addressed the reviewer's comments. A point to point basis reply to the reviewer's comments is below for consideration.

This revised version of the manuscript is much improved with respect to the first version, thanks to the authors for addressing the point I raised in my review. I am happy with the changes, I just ask for some minor edits just to avoid confusion in the definition of the model product used.

Although it has been generally better clarified, in the text there are still a couple of points where it is not clear where the mass mixing ratio data comes from. It is always mentioned the ECMWF IFS, but this might create confusion because it could suggest that ECMWF forecasts provide aerosol information, which it is not the case. Here below the few corrections needed.

line 32-35,p3: attention, confusion about what type of boundary conditions IFS is used for. I suggest to rephrase similar to "In this study, for the first time a comparison of aerosol profiles provided by the Copernicus Atmosphere Monitoring System (CAMS) with long term measurements of the ceilometer network measurements operated by the German weather service Weather Service (DWD) is presented. CAMS forecasts are quite relevant as it is often used to provide boundary conditions for regional chemistry (?) models.

Response: We have rephrased the sentence (page 3, line 24-28).

line 5,p5: "[...] of the original CAMS model output used in this study [...]"

Response: We have revised the sentence (page 4, line 27).

line 19-20,p28: . "We have compared aerosol model simulation of the European Centre for Medium-Range Weather Forecast Integrated Forecast System (ECMWF-IFS) provided by CAMS"

Response: We have revised the sentence (page 26, line 24-26).

[revised manuscript text omitted]